# Entropy and the Experience of Heat

**DOI:** 10.3390/e24050646

**Published:** 2022-05-04

**Authors:** Hans U. Fuchs, Michele D’Anna, Federico Corni

**Affiliations:** 1Center for Narrative in Science, 8400 Winterthur, Switzerland; hans.fuchs@narrativescience.org; 2Faculty of Education, Free University of Bozen-Bolzano, 39042 Bressanone, Italy; 3Liceo Cantonale Locarno, 6600 Locarno, Switzerland

**Keywords:** entropy, extensive quantity of heat, phenomenology, conceptualization, early history of thermal physics, teaching and learning of thermodynamics

## Abstract

We discuss how to construct a direct and experientially natural path to entropy as a extensive quantity of a macroscopic theory of thermal systems and processes. The scientific aspects of this approach are based upon continuum thermodynamics. We ask what the roots of an experientially natural approach might be—to this end we investigate and describe in some detail (a) how humans experience and conceptualize an extensive thermal quantity (i.e., an amount of heat), and (b) how this concept evolved during the early development of the science of thermal phenomena (beginning with the Experimenters of the Accademia del Cimento and ending with Sadi Carnot). We show that a direct approach to entropy, as the extensive quantity of models of thermal systems and processes, is possible and how it can be applied to the teaching of thermodynamics for various audiences.

## 1. Introduction

In this paper, we sketch answers to the question of where we might find the roots of an *Experientially Natural form of Thermodynamics* (ENT: Table 1; [1])—where *entropy* emerges directly and easily as the *Extensive Thermal Quantity* (*ETQ*) or *Extensive Quantity of Heat* (*EQH*) (i.e., as the *amount of heat* in thermal processes). We accept continuum thermodynamics as the scientific basis, and seek answers by considering embodied cognitive science, the early history of thermal science and engineering, and modern applications in the field of thermal processes. The results of these considerations will be compared with a modern version of a *Dynamical Theory of Heat* (DTH; see [2]); this allows us to argue that *entropy* resembles an imaginatively constructed *EQH*.

By *experientially natural* we mean an approach where the concepts and relations used to structure formal models and a theory of thermal processes result from basic *mental acts of schematizing/abstracting* [3] and *imagining*—arising in interactions between physical and linguistic experiences—which are fundamentally human thought processes [4,5,6,7,8,9]. We shall see that advances in embodied cognitive science in general, and cognitive linguistics [9,10,11,12,13] and narratology [14,15] in particular, can shed light upon this issue [16,17,18,19,20,21,22,23]. Moreover, the history of early work on thermal phenomena from the *Accademia del Cimento* (Magalotti, 1667 [24]), up to and including Sadi Carnot’s *Réflexions sur la puissance motrice du feu*… (Carnot, 1824 [25]), and the recent construction of a theory of the dynamics of thermal systems based upon continuum thermodynamics [2,26,27,28] can be used as guides for creating a form of thermodynamics where temperature is the thermal potential and entropy is the fundamental *extensive property* (*ETQ* or *EQH*) that is *transferred* and *produced* [29] in thermal processes. This will be our answer to the challenge of creating an ENT.

### 1.1. Heat as a Force of Nature

We shall argue that human experience presents *Heat* [30] to us as a *Force of Nature* (FoN, [31]; see [8,32,33,34]). FoN refers to a pervasive, experiential, gestalt-having intensity, which, by extension, has power as its fundamental quality. Using the term *Heat*, and capitalizing the word as we do for this gestalt, means that it refers to the totality of the experience, and not what a physicist would call (quantity of) *heat*, denoted by the letter *Q*. By engaging with thermal phenomena both physically (through direct physical encounters) and narratively (by communicating our physical experiences), basic schemas/abstractions (such as thermal intensity/level, tension, and slope/gradient; amount of heat as fluidlike entity, flow, and production of amount of heat, and storage of amount of heat; power of heat; process, agency, cause, etc.) arise in our mind. The products of such mental activity are the building blocks of formal models of thermal processes. Moreover, experiencing and imagining the interaction of *Heat* with other *Forces of Nature* introduces us to the notion of the *Power of Heat*.

We contrast such an experientially natural form of thermal models with *Traditional Equilibrium Thermodynamics* (TET; see, in particular, [35]). There are at least two aspects of TET that could be deemed cognitively unnatural: (1) in ENT, we accept that *processes* rather than states are products of experience; TET, on the other hand, is a theory of the *equilibrium of heat*—meaning it is not a dynamical theory (in particular, there are no initial value problems to be formulated [36]; see [35,37]); (2) in ENT, we directly conceptualize entropy as the extensive quantity of thermal processes; since, in TET, “quantity of heat” (i.e., energy transferred in heating and cooling) is not an extensive quantity (see below), and thus, we do not have direct access to the notion of a measure of *extension* of *Heat*. This measure needs to be constructed indirectly and formally (Clausius, 1865 [38]), and is therefore hard to recognize as the *ETQ* [39,40].

In this paper, we shall be concerned mainly with the second of these aspects. This allows us to reformulate the initial question as follows: how can a sense of the extensive thermal quantity (*ETQ*) (i.e., entropy) be developed as directly and as painlessly as possible? Here is the short answer: it will be developed as the *extensive aspect* [41] of the experience of *Heat* as a *Force of Nature*.

### 1.2. Extensive Thermal Quantity

Before we proceed, we should clarify more precisely what we mean by *Extensive Thermal Quantity* (*ETQ*) or *Extensive Quantity of Heat* (*EQH*; note that the capitalized *Heat* refers to the FoN, the totality of thermal experience) and describe why *heat* in TET is *not* an extensive quantity. An extensive quantity in theories of *Continuum Physics* (CP; [26,27,28]), or in the mathematically simpler theories of spatially *Uniform Dynamical Systems* (UDS; see [2]), is a quantity [42] for which we write a *law of balance* relating to (the rate of change of) the *amount* of this quantity found in a region of space, in order to *transport* it across the surface of this region, to *source rates* and to *production rates* in the region. If we use the symbol C for the value of the *EQH* stored inside a region [43,44], $I_{C}$ for transports, C, $\Sigma_{C}$ for source rates, and $\Pi_{C}$ for production rates [45], we have:(1)$$\dot{C}=I_{C,net}+\Sigma_{C}+\Pi_{C}$$
(for symbols, see Table 2). In other words, it is assumed that, in general, the following quantities associated with the extensive quantity exist and are related by a law of balance: a *density*, *conductive* and *convective current densities*, a *source rate density* (i.e., a quantifying radiative transfer, such as the interaction of a body [46] and a field), and a *production/destruction rate density* (see [2,47,48,49]). In the case of heat, the production rate will be non-negative. In other cases, other quantities are strictly equal to zero (such as a radiative term for electric charge).

In summary, an extensive quantity such as electric charge, momentum, angular momentum, entropy, or amount of substance is one for which we can formulate a law of balance whose general form for UDS is given in Equation (1); whether the quantity is conserved is of no concern in this classification. It should now be clear that there is no *heat content* in TET, and there are no *convective* and *radiative* transfers of (quantities of) *heat* either [50]. The only permitted usage of the concept of (quantity of) *heat* in TET is for a quantity of energy transferred (into or out of a system) due to a temperature difference (which is equivalent to saying that the transfer is *conductive*: it is the type of energy transfer facilitated by conductive currents of entropy). In other words, it is impossible to formulate a law of balance for (quantity of) *heat* in TET—*heat*, as it appears in our textbooks, is not an *ETQ*!

### 1.3. Outline

Here is a brief outline of the structure of our paper. In Section 2, we describe aspects of the phenomenology of *Heat* as a FoN and how images of an *ETQ (in the form of an EQH)* may arise in our mind, which will serve us well in thinking about, and modeling, thermal processes. Most importantly, human experience provides us with perceptual units we call *Forces of Nature* (FoN)—*Heat* is but one of these; others are called *Fluids*, *Electricity*, and *Motion*. It turns out that one of the primary aspects of FoN is their *extension*, which, in the case of *Heat*, is invisible [51] and not so easily directly perceived—hence our question, once again: how can a sense of the *Extensive Thermal Quantity* (*ETQ*) arise and take form as directly and as transparently as possible?

We know, in hindsight, what imagery is required for us to be in possession of an *ETQ*: we need to imagine a fluidlike quantity that is contained in materials, flows in and out, and can be produced in irreversible processes. Furthermore, we need to accept intuitively that this quantity, upon entering air (and other objects), makes the material warmer and lets it expand. Finally, we need to be able to imagine how this quantity enters the coupling of different forces of nature, such as when *Heat* produces *Motion*, or *Electricity* and *Heat* enter into a joint activity in thermoelectric devices [52].

If we wish to accept such an image—a quantity imagined as a kind of non-conserved quasi-fluid—capable of being produced (having a *non-negative production rate*) in irreversible processes—we must reconsider simplistic ideas of how experience and our minds work ([53,54,55,56,57,58,59,60,61,62,63,64,65,66,67,68] and Section 2.3). We need to shed superficial assumptions about the materiality of *amounts of heat* (or, for that matter, amounts of electricity, fluids, or gravitational mass, and all other extensive quantities of modern physics as well). We need to accept that most basic aspects of science are products of the imagination working on schematic products of experience [8,9,13,21,22,23,32,69] and are pervaded by figurative thought [9,10,11,12,13]. Elements of an embodied model of cognition, described in Section 2, will help us recognize how imaginative acts let us come up with what is required in our case. We shall see that abstract schemas applied in metaphoric projections and analogies are prevalent in everyday language and science, and are used to communicate our experiences of thermal phenomena [69]. Such figurative elements and processes of our understanding can, when formalized, be used for constructing a formal theory of *Heat* as a *Force of Nature* (see Section 4).

Here is some more detail concerning these important cognitive issues. Our goal of creating an *Experientially Natural form of Thermodynamics* (ENT) clashes with much of what is commonly held true in cognitive science concerned with the origin and (developmental) change of concepts used in representing the (natural) world. Traditional approaches to these fields—basically (Scientific) *Theory Theory* (TT and STT) and *Simulation Theory* (ST)—take their starting point in TT and ST models of a Theory of Mind (ToM or Folk Psychology), i.e., in models that are supposed to tell us how we learn to understand other humans (other minds). These models are transferred to science learning leading to an important product: *Conceptual Change Research* [54]. Take TT which maintains that our cognitive abilities are the result “of a body of implicit knowledge, constituted by a representational system in the form of law-like generalizations or innate mechanisms [55]”. STT goes one step further and claims that our understanding of other minds develops through theory-forming abilities we know from the natural sciences.

If we transfer this view to how a child learns to understand nature and creates concepts (as stable small-scale units of cognitive structure), and if we take for granted that our “naïve” or common-sense concepts will certainly be wrong and misleading, the need for robust *Conceptual Change* is demonstrated. However, there exist rival approaches to learning and understanding science that acknowledge *productive resources* (PR) of learners [56]. PRs may be small-scale knowledge structures (for instance, *phenomenological primitives* such as *blocking* and *balancing* [57]) or cognitive and methodological abilities (such as *analogical reasoning* or *refining* of “raw intuitions”). In this framework, learning is understood more as acts of refinement, differentiation, and placing in context of “raw intuitions” than as change or replacement of “faulty” concepts [58].

In this paper, we take an approach strongly resembling the latter: We all have access to cognitive tools and specific cognitive elements—image schemas, metaphors, the capacity of analogical reasoning, and narrative understanding—forming through organism-environment couplings that can serve us well in creating imaginative forms of understanding the natural world and create concepts, models, and theories [59,60,61]. Using the idea of PRs, we may call FoN a medium-scale PR that takes its power from its structuring as a metaphoric web and how it is embedded in narratives (both in stories for young learners and more formal explanatory narratives used by scientists and engineers [8,62,63]). Moreover, as a narrative element, FoN are a PR for the important activity of (formal and informal) modeling and simulation [64].

If we take this more “organic” and “evolutionary” view, we can turn the tables and ask if we do not learn to understand other minds similarly to how we get to know nature and interpret our physical experience. A narrative approach to learning about ToM that bears some resemblance to our narrative approach to learning about Forces of Nature is the Narrative Practice Hypothesis developed by Hutto and Gallagher [65,66,67,68].

To be clear, all of this does not mean that learning science will be easy, and nothing can go wrong (or no “conceptual change” will ever be needed). What this means is that the role of basic embodied (figurative, imaginative) structures of understanding that can be made use of in science learning is different from what is commonly assumed. Thermodynamics, particularly the issue of the *EQH*, is a case in point.

Section 3 is devoted to a description of certain aspects of early investigations into thermal phenomena that are important for approaching the central question raised in this paper. We start with the Experimenters of the *Accademia del Cimento* and move, via Black, Fourier, and Laplace, to Carnot (roughly from 1660 to 1830). We show how, upon examination of these investigators, schematization of direct experience evolved into a *model of caloric* as an *ETQ*, called the *Caloric Theory of Heat* (CTH) [70]. With the help of this model, results were produced that still hold today. The most important of these are theories of adiabatic processes and the speed of sound in fluids; Carnot’s expression for the power of thermal engines is analogous to the power of a waterfall and Fourier’s theory of pure heat conduction. As is well known, despite such fundamentally important results, the CTH did not survive. Even though this will take us beyond the period we consider here, to about 1865, we shall briefly discuss what led to this development in the history of thermal physics (see Section 3.5).

If we compare the images of *Heat* arising from experience (Section 2), and let our imagination be guided by the positive results of the early scientific investigations of thermal phenomena (Section 3), we can, as we have already stated, construct a theory of *Heat* as a *Force of Nature* with the *Extensive Thermal Quantity* (*ETQ*) or *Extensive Quantity of Heat* (*EQH*) as a primitive concept (Section 4). If we compare such a formalism with thermodynamics in UDS or CP, we see that we can understand *entropy* as the *EQH* constructed it.

In Section 5, we shall describe a small number of models that apply this *direct entropic approach* [52] to thermal phenomena and discuss their application to the teaching of thermodynamics in high school and university courses. The final section is used to summarize our results.

## 2. Experiencing and Communicating about *Heat*

Let us collect and describe a few everyday thermal phenomena. Our main focus will concern how the phenomena are experienced and then talked about. Note that a description of experience will immediately involve an act of communication and conceptualization—experiencing and communicating form a unified activity [4]. It does not make sense to assume that our words, which describe what we experience, are just a neutral reflection of the act of perceiving. We should much rather believe that our words shape our experience as much as the physical process involved in perceiving does—our words will be part of the experience [71,72,73].

Therefore, after describing examples of thermal phenomena, we shall reflect upon the form of narrating such experiences from an embodied mind perspective. Embodied approaches to cognition are models from which theories of *schematic forms*, *metaphor*, *analogy*, and *narrative* have flowed. Using these theories as tools, we shall be able to show that the mental structures needed for creating formal approaches to a science of heat exist in everyday communication. In particular, we will be able to trace the roots of our experience with an *EQH*.

### 2.1. Experiencing Heat

Experiencing thermal phenomena starts with the perception of *hot* and *cold*—and every other degree on the scale of *hotness* (or coldness). During the day, temperatures go up and down. If we put water or food on a fire, they become warmer. If we let our coffee sit on the table, it gets cooler until it has turned as cool as the environment; we notice that it will get cooler more slowly if it is in a well-insulated cup with a lid. Heated stones thrown in cold water get cooler and make the water warmer. If we step from a place in the direct sun into the shade under a tree, we feel how much cooler it gets over a distance of just a few meters. If we sit in front of a fire in an otherwise cold environment, we may be hot in front and cold on our back.

*Degree of hotness* is felt as an *intensity* during a thermal experience. Differences of hot and cold are experienced as *tensions*, and they are imagined as *level differences* in an imaginary landscape. We are told that “[…] bodies of water […] differ in their temperatures, providing a thermal ‘landscape’ which might serve as an orienting cue in fish migration” [74], p. 152.

During a thermal experience, sensing hot and cold is the immediate activity of an organism; however, there is more to such phenomena, which are barely a step away from directly sensing an intensity on the hotness scale. Thermal phenomena arise in chains of processes in and around us. Sunshine makes a hillside hot in summer, the hot ground heats the air over it; it will expand, get lighter, rise, and take its humidity with it so clouds can form in the sky. We burn fuel in the cylinders of a combustion engine to produce heat which, in turn, will make the gas expand and move the pistons. A thermal process is caused, and this, in turn, causes other processes—something we observe quite directly. In this respect, thermal phenomena are very much like other dynamic physical processes, such as those related to electricity, fluids, gravity, or motion. In sum, we experience dynamics in nature which we describe—and at the same time interpret—as chains of interacting phenomena. 

The phenomena we observe may be gentle, strong, or outright violent; they may happen slowly or quickly. In this, we recognize the power of these phenomena (i.e., the *power* of processes interacting with and causing other processes).

So far, we have described two generic aspects of thermal phenomena: *hotness*, which we quantify as temperature, and *power*, which is measured by its causal effect upon other processes. With one exception, we have avoided using the word *heat* in narrating the examples above. It is time we turn to this notion and discuss what we might mean by it; however, before we get to this point, we should talk about some thermal phenomena, where using the noun *heat*—or, as we shall see, *cold* as well—is quite common and apt.

Imagine you are standing with your bare feet in a cold stream. If you stay in the water long enough, you will feel “something creeping up” from your feet into your legs making them ache. It is not just that parts of your legs get gradually colder: the parts of your legs that are lower down cool down before the parts higher up. Our mind readily creates the image of “something creeping up” which we call *cold*. A similar situation is experienced if we stick a long metal spoon in hot tea: something, which we call *heat*, quickly “flows up” along the handle of the spoon. Imagine further you are in a home in deep winter, and you know, from experience, that *cold* “finds its way” through walls and windows and cracks between walls and windows. You also know that you need to light a fire so the heat it produces can fight and balance the cold.

There is still more to the story of *heat*. Consider using a battery for operating some electrical appliance. Interestingly, the battery will get warm. Now, if the battery is rechargeable, it will get warm during recharging as well. We say that, in the former case, chemical reactions make electricity flow; at the same time, *heat is produced*. In the latter case, electricity makes the original chemicals reappear in the battery; and again, heat is produced. We have reversed the chemical and electrical processes when going from the former to the latter operation, but we obviously could not reverse heat production. Apparently, heat can “go away” if it is “let go” to colder places, but it cannot disappear in the full sense of the word—it cannot be destroyed.

Think about it: we would be at a great loss if we could not describe such situations in terms of *heat* (and sometimes *cold*) flowing in, out, and through bodies; residing in these bodies; and being produced in a fire and some other processes. If, instead, all we could do is speak about the temperature changing locally and in the course of time, our narrations would become more than just awkward. Simply imagine what it would be like if our mind did not provide us with an image of an invisible entity that flows into and out of bodies, and when it is inside them, makes them warm (or cold in the case of *cold*). In fact, would we even have the word *heat* if we did not have the experience and image of heat as “some stuff?” 

We shall accept the imagery of some “thermal stuff” in the sense described in these examples, as the third primary aspect of the experience of thermal phenomena, in addition to intensity and power. Clearly, if there is some “stuff,” there is more or less of it, depending upon circumstances and time. This tells us that we associate an *extensive aspect* with thermal phenomena. The most fitting linguistic term that can and should be used to name this “stuff” will obviously be *heat*, or if we want to be a little more formal, *quantity of heat*.

### 2.2. Heat as a Force of Nature

These are the opening lines of Sadi Canot’s book, *Réflexions sur la puissance motrice du feu*, of 1824 [25] (pp. 1,2): “[w]e are all aware that heat can be the cause of movement, that it even possesses great motive power: the steam engines, now so ubiquitous, are a proof that speaks to anyone who can see. 

“It is to heat that we must attribute the great movements which attract our attention here on Earth; it is to heat that we owe the agitations of the atmosphere, the rise of clouds, the fall of rain and other meteors, the currents of water which channel the surface of the globe and of which man has succeeded in using but a small part for his own purposes; finally, earthquakes and volcanic eruptions also recognize heat as their cause.

“It is from this immense reservoir that we can draw the moving force necessary for our needs; nature, by offering us fuel everywhere, has given us the faculty, at all times and in all places, of giving birth to heat and to the power which results from it. To develop this power, to appropriate it to our use, such is the object of heat engines”[75].

*Heat*, (*chaleur* in Carnot’s text quoted here) is here not so much a term for *quantity of heat*—as we have used the term above when we described some phenomena—but rather a word for what we have called the unified experience of *thermal phenomena*. Carnot’s lines describe *Heat* [30] as a power, a *Force*, or an agent.

Modern cognitive science tells us that we can, and readily do, experience simple *perceptual units* or gestalts in complex scenes. *Heat* is one such gestalt if we speak of it in the way Carnot did. When we use the term *Heat* in this manner, we refer to something unified and simple, and we always recognize for what it is—we always know when we have a thermal experience. However, *Heat* can be analyzed: it shows us primary aspects, and to be certain, upon further analysis, less primary ones as well. As we can ascertain from the descriptions in Section 2.1, the most schematic or abstract of these aspects are *intensity*, *power*, and *extension*. We shall call any perceptual unit that shares these schematic features with *Heat*, a *Force of Nature*.

Phenomena such as *Wind*, *Rain*, *Fire*, *Ice*, *Water*, *Air*, *Humidity*, *Electricity*, *Fluids*, *Gravity*, and *Motion*, are members of the same family. Just compare *Wind* with *Electricity*. *Wind* can be more or less intense (differences in its intensity are felt as tension), it is extended in space (usually over a vertical surface through which it blows) and can be more or less powerful. The same schematic aspects are found in *Electricity*: the phenomenon presents itself to us through tensions (differences of its potential), an extensive aspect (i.e., a quantity of electricity such as electrical charge), and power. In each case, they appear to us as experientially primitive figures or shapes whose basic schematic aspects are intensity, extension, and power. In macroscopic physics, the *Forces of Nature* we deal with are treated in the following fields of science: *Fluids*, *Electricity*, *Magnetism*, *Heat*, *Gravitation*, *Chemical Substances*, and *Translational* and *Rotational Motion*. [2,27].

In our imaginations, *Forces of Nature* are experienced as agents [76,77] or patients [78], and that is how we speak about them. They are active, they do things, or things are being done to them. An agent can be *big* or *small*, *relaxed*, or *tensed*, and *more or less powerful* (here, we recognize the three basic aspects of a *Force of Nature*). As agents, *Forces* appear quite commonly in our stories of natural phenomena and processes. For a child, this makes them partners with natural encounters; for an expert, it makes them figures whose actions we can reason about when we are engaged in modeling of systems and processes (see [79]).

### 2.3. The Nature of Experience and Conceptualization

A few words concerning experience and conceptualization are in order. We might not generally think that philosophical and cognitive issues have much bearing upon the construction, use, and understanding of a science—natural science follows the evidence presented to us directly. We typically think that we do not need sciences of the mind telling us what is true regarding the constructs found in the natural sciences.

This attitude is precisely the problem, and particularly so in thermodynamics. How we conceptualize our experience is not the result of an objective state of affairs in the world that projects onto an accepting (dis-embodied, immaterial, transcendental) mind that delivers and manipulates propositions we must take literally. Models and theories in physics are, rather, the product of imaginings following the interaction of several *forms of experience*. For our purpose, the most important of these are (1) *direct physical experience* (i.e., the interaction of a human organism with the physical world), and (2) *linguistic experience* (i.e., linguistic interaction (in the broadest sense of linguistic) between several humans jointly attending the same physical scene). 

Understanding *experience* as resulting from organism–environment interactions (i.e., from action–perception feedback loops), is a fundamental assumption in much of modern cognitive science [5,6,7,11]. In sum, what is important to us is linguistic evidence and that the abstractions we use in (macroscopic) physics derive from our sensorimotor (i.e., bodily) interactions. We derive schematic elements from our experiences that are used and re-used imaginatively in the construction of figurative (metaphoric and analogical) conceptual webs (Fuchs et al. forthcoming [80]). In other words, concepts, models, and theories are figurative affairs. Let us see, by way of example, what these structures are.

### 2.4. Abstraction through Schematizing Actions of the Mind

Experiences leave *traces* [81] that our imagination can work upon. Some of these traces are highly abstract *schematic structures* (rather than richly complex concrete pictures and memories). Our sensorimotor interactions with the natural world most frequently concern spatiotemporal perception, the experience of the motion of bodies (which includes, most importantly, how it feels when we move our bodies), and the feel and behavior of fluids. We shall see that our language is full of figurative elements derived from this type of experience; we use these elements not just when we deal with concerns of everyday life, but when we work with scientific affairs as well.

Cognitive linguistics has taught us about some of the abstract schematic structures of recurring sensorimotor experiences: so-called *image schemas* [82,83] such as scale, path, polarity, tension, balance, motion, process, up-down, front-back, container, in-out, substance, flow, force (including a number of *force dynamic schemas* such as compulsion, making, letting, enabling, and opposing), and so on. What we call *concepts*, particularly in the sciences, result, by and large, from *metaphorically projecting* such schemas upon our rich and concrete experiences. If we say, for instance, that in an argument, we would like to *go a step further*, we project the path schema (and whatever else it entails, such as *making progress along a path*) upon the experience of an *argument* [84]. Alternatively, take an example from a description of a natural phenomenon: *cold easily found its way through the cracks in the wall*. Apart from the path schema, we find at least the schemas of flow, fluid substance, and enabling in this expression. Such use of schematic forms and their projections creates much of our *figurative* understanding of the world around us.

In sum, we look upon conceptualizations in physics as resulting from the activity of a mind that works figuratively to a profound extent. In the following subsections, we shall see what kind of figurative structures are used to create and structure a science such as macroscopic thermodynamics.

### 2.5. Metaphors and Metaphoric Webs for Communicating about Heat

If we collect statements about *Heat* as a *Force of Nature*, made in everyday, technical, and scientific communication, we come across many of the image schemas listed in Section 2.4. They are applied in so-called *conceptual metaphors*, meaning that whatever knowledge of the physical world they reflect, it is projected onto new experiences such as the action of *Heat* in nature and machines. A very simple and frequently occurring example is “*the temperature is rising quickly*”. Clearly, this statement cannot be meant literally—temperature is neither “high” nor “low,” nor is it “rising” or “falling,” nor is it rising “quickly” (in the literal sense of these words). In other words, the statement is meant figuratively—it applies the schematic (abstract) images associated with the concrete experience, and thus, these words apply to a different example of experience. This type of projection of schemas is called *metaphoric* [85,86,87,88].

If we listen to people speaking, or scour the Internet for expressions about *Heat*, we find countless examples containing elements and constructs such as *store*, *hold*, *lack of*, *collect*, *flow*, *heat moves*, *heat causes*, *heat makes*, *heat counteracts*, *pump heat*, *force heat*, *block* (*hold back*) *heat*, *enable heat*, *prevent heat*, *produce heat*, *heat is a level*, *heat is a location or landscape*, *balance of heat and cold*, *thermal tension*. Here are some examples (all the expressions can be found online): Global Ocean **Heat Content** 1955-present 0–2000 m.…the object serving merely as a **container for heat**.**Heat flows** ‘downhill’. It flows from a locality of **high temperature** to a locality of low temperature, irrespective of the **heat content** in each locality.Modelling the response of lizards to **thermal landscape**.There needs to be **tension between hot and cold**, so that pinot noir can ripen slowly and show a true and exciting expression of the wine.Geothermal add-ons for heat pumps on the market today **collect heat** from the air or the ground.To reverse the process, (so that **heat flows uphill** from a cold reservoir to a hotter reservoir), one must put in additional external energy to **“pump” heat** from the…Modeling Heat Movement. **Heat moves** from one place to another in three ways…New research, however, shows plate dynamics are driven significantly by the additional **force of heat** drawn from the Earth’s core.… in Dallas, the Texas **heat is a force** to be reckoned with.(R. Clausius) Law of the dependence of the active **force of heat** upon the tempera.…consequently, **heat is an agent** which is competent for the consolidation of strata, which water alone is not [89] (J. Hutton, 1795).**Internal Heat Drives** Jupiter’s Giant Storm Eruption…The cold is injurious to the blood, but dry **heat counteracts** the cold.Solar **heat lets** you spend more time in your pool……use electricity to **make heat go** where it **does not want to**…The reflective nature of the foil will **prevent heat from disappearing**…

Analysts in cognitive linguistics would say that we humans have access to a number of conceptual metaphors that together form a conceptual web with which we understand thermal phenomena [80]. The most general metaphor relating to our subject could be expressed as heat is a powerful agent (i.e., heat is a force of nature). There are sub-metaphors such as heat is a fluid substance, objects are containers for heat, temperature is a level, heat forms a landscape, and heat is a fluid under tension.

We argue that such metaphoric webs underlie the understanding we have of experiences of the many phenomena we are exposed to. As the examples show, it is not just in innocent everyday conversation that we speak like this; engineers and scientists use the exact same language when they communicate about phenomena such as *Heat*. We obviously have access to a form of knowledge that lets itself be formalized in macroscopic theories of heat.

This brings us to an important point concerning metaphorical understanding of thermal processes. There is a metaphor we might express as heat is the motion of the smallest parts of bodies (Clausius). It, and whatever else it entails, forms a metaphoric web just like heat is a force of nature. It is important to understand that “large-scale” metaphors such as these are largely incommensurable. They each give us a fundamentally different perspective on the same field of experience. Each perspective creates its own images, figures, metaphors, and analogies (on analogy, see Section 2.6), and we cannot simply reduce one to the other.

### 2.6. Form and Role of Analogical Reasoning

The image schemas referred to in the previous paragraphs are so basic and fundamentally important that it should not come as a surprise that they appear again and again in our experience of different *Forces of Nature*. We do not need to have completely different images when we metaphorize different phenomena. We reuse a few basic schemas and their metaphoric projections when we communicate about and create models of various *Forces of Nature* that are active in natural and technical systems. This imaginative activity makes the fundamentally different phenomena—*Fluids* and *Heat*, *Electricity* and *Motion*, and so on—similar in our mind [33,90,91,92,93,94]. This similarity allows us to consider the various *Forces of Nature* as being (largely) analogous. Note that analogy is not an objective feature of the world out there; it is a product of human imaginings.

Of all the examples of experience that lead to schematic structures (Section 2.4), let us just consider those that originate in our interactions with fluids. We have good reason to believe that the domain of fluids furnishes schemas that are then used to metaphorically understand (aspects of) FoN, such as *Heat*, *Electricity*, *Chemicals*, and *Motion*. If we now inspect the formal theories of these phenomena presented to us in *Continuum Physics* [27], or the *Physics of Uniform Dynamical Systems* [2], we find analogous laws of balance used for dealing with amount of fluid, amount of electricity (charge), amount of heat (entropy), amount of substance, amount of motion (momentum and angular momentum), and gravitational mass. For all their similarity, we can call the formal concepts *fluidlike quantities*; therefore, it is perfectly alright to say things such as *heat is like water*, *momentum behaves like an electric charge*, *an electric current is like a current of water*, *a capacitor is like a vessel*, and so on. Note, these mappings do not represent metaphors; they are analogies. Where metaphors are unidirectional mappings between two domains (we can say that heat is a powerful agent but not a powerful agent is heat), Genter’s model of analogy as a form of structure mapping [90]—where (some) elements of a domain A are mapped upon (some) elements of a second domain B—creates a structural similarity between these domains; this, in turn, suggests that B can be compared with A, but also that A can be compared with B (in other words, there is a partial bidirectionality). We can (partially) compare *Heat* with *Electricity*, but also *Electricity* with *Heat*. If we feel more secure in our understanding of one of these domains, we can transfer (certain aspects of) our knowledge analogically to the other—it does not matter which is which (however, as a result of concrete experience, we are often left with a certain asymmetry between the domains that we take as analogical—we do prefer one over the other).

This kind of transfer has been used in the construction of theories and models, and it has proved an invaluable tool in teaching physics (see, in particular, [2,33,49,91,92,93,94,95,96]). For us, the most important message is this: if there are extensive quantities that are needed in our theories of motion, electricity, fluids, chemical processes, and so on, there certainly needs to be an analogous extensive quantity, which is *ETQ* or *EQH*, in the field of theories of thermal phenomena. What twisted form of physics we end up with if we want to forgo the *EQH* and equate heat with (some aspect of) energy rather than using analogical reasoning as here discussed, has been described in [39] as a *Surrealistic Tale of Electricity*.

## 3. From the Accademia del Cimento to Sadi Carnot

If it is accepted that schematizing and imaginative activities of our mind, as expressed in figurative language, give us access to a notion of an *Extensive Quantity of Heat* (*EQH*), we should wonder why this concept has disappeared from TET. We shall see that some forms of a notion of an *EQH* were very active in early studies of thermal phenomena—the most famous of these was the *Caloric Theory of Heat* (CTH; [97,98]). There are different reasons why, ultimately, it has been said to be unsuccessful; we shall briefly discuss some of the reasons at the end of this section.

First, however, we shall describe imaginative aspects of early investigations into thermal phenomena [99,100]. With the exception of some aspects of what follows for a formal theory of thermodynamics from Laplace’s and Carnot’s work, our focus will be on qualitative (linguistic) formulations that show how the notion of an *EQH* took shape and was used in developing early formal approaches to a science of heat [101,102]. We shall limit our search for understanding this concept from the time of the *Accademia del Cimento* (about 1660) to that of Sadi Carnot (around 1825). Importantly, given the varying and often conflicting special assumptions made concerning an *EQH* during that period, we want to know what basic schematic features of this concept were shared by the investigators. Answers to this question may serve as pointers to a modern form of thermodynamics based upon a direct approach to an *EQH*.

### 3.1. The Experimenters of the ADC: Cold as a Force of Nature

In the second half of the 17th century, a group of Florentine scientists, the Experimenters of the *Accademia del Cimento*, came together; they were determined to advance the experimental method introduced by Galileo as the basis for scientific investigation. Their investigations were published as *Saggi di naturali…* in 1667 [24] and translated into English soon after by Richard Waller in 1684. In this book, they report on a multitude of observations and experiments. In particular, on pages CXXVII–CLXV of the original publication, they describe experiments concerning the artificial freezing of various liquids in a glass bulb, featuring a long thin neck for measuring volumes, stuck in a container filled with a very cold mixture of ice and salt (Figure 1). They were interested in exploring the power of the *Force of Cold* by observing by *how much* and *how quickly* the liquids would change their volumes upon being exposed to the cold environment. They recorded the times with the help of a pendulum for which they counted the number of “vibrations” (see the table on the top right of Figure 1, column 5) in order to detect changes to the level of the liquid in the neck (columns 1 and 2), and for alcohol (their “thermometer,” see columns 3 and 4). 

Note that, by observing how quickly changes occur, they treated the actions of *Cold* upon the volume of liquids as a matter of *dynamics*. Data collected in the diagram of Figure 1 demonstrate the typical, exponentially decaying, cooling curve of a simple body. It seems that this was likely to be the first and last occasion when *time* appeared in a study of thermal physics—with the notable exception of Fourier’s theory of the conduction of heat—until modern thermodynamics was conceived of as a unified approach to thermodynamics, heat transfer in continuum physics, and the physics of dynamical systems [2,26,27,28].

From the descriptions in the *Saggi*, we can get a picture of the conceptual framework with which the Experimenters tried to interpret the observed phenomena. In the following quotations, using our translations, the page numbers are those of the original document [24]; the original Old Italian texts are given in [103]. We shall see, by quoting just a few lines, that they had a general notion of *Cold* (and *Heat*) as a *Force of Nature* whose primary aspects are *intensity*, *quantity*, and *power*.

*Heat and cold as fluidlike substances*: (pp. 127,128) “…while it is considered that where the fire melted into very fast sparks, chasing itself through the thickest crevices of the stones, and of the metals themselves, it opens them, liquefies them, and reduces them into water…”; (p. 150) “…and that this separation did not begin until after the water had begun to take the extremely strong cold”. Even though they used *Cold* as an important concept in their interpretations, they were unsure if *cold* should be thought of as a lack of *heat* (pp. 128,129): “[t]he reason for the freeze has been speculated about in different ways at all times, whether this really arose from a proper and real substance of the cold… or whether the cold was nothing more than a total deprivation, and expulsion of heat”. 

*Heat and cold have a quality, intensity; there is a degree of heat and cold*: (p. 154) “…in order to see with the thermometer, with what degrees of coldness…”; (p. 194) “…to enroll in them the various regrowths, which are operated by different degrees of heat…”

*The power of heat and cold,* i.e., *heat and cold as agents capable of producing phenomena*: (p. 11) “…so that the water has room to thin out when the heat of the season forces it to do so” (pp. 127,128). “Indeed (which causes us more amazement) we see the cold working with such a violent force in the freezing of fluids…”

The Experimenters were anything but certain about their judgements concerning conceptualization of heat (and cold). Nevertheless, their language betrays the typical imagery we create when confronting our experience of *Forces* such as *Heat* and *Cold*—our mind creates a perceptual gestalt demonstrating characteristics of intensity, extension, and power. The same observation has been made in a study of the *Saggi* by Wieser and Carey [104]. As we have done here, they concentrated on the experiments of artificial freezing. Here is their summary of the state of conceptualization by the Experimenters: “[t]he Experimenters’ concept of heat had three aspects: substance (particles), quality (hotness) and force” ([104] p. 289).

Nevertheless, there is much that can be criticized in the *Saggi* if we take a modern scientific perspective. The Experimenters’ power of analysis was certainly not up to task. Using the alcohol thermometer in parallel with the bulb for the liquid they studied must strike us as strange; however, that misses the important point that the Experimenters were searching for a dynamical description of thermal phenomena. We can understand their use of the thermometer as an instrument showing the *Force* of *Cold* acting out of, or emanating from, the ice–salt mixture as it affected the alcohol in the thermometer over the course of time. Their basic aim—looking for the *Forces* of *Heat* and *Cold*—was the same that guided Carnot’s thinking.

### 3.2. Joseph Black: Temperature and Quantity of Heat

In his lectures from the second half of the 18th century, published posthumously in 1803, Joseph Black [105] showed in a precise, clear, and explicit manner how to distinguish between temperature and quantities of heat possessed by bodies (i.e., between the intensive and extensive aspects of *Heat*). He introduced concepts of *specific heat* or *heat capacity* [106] and *latent heat* (the latter limited to processes of phase change). His conceptualizations are a clear example of what later became known as the CTH.

Black’s work and (linguistic) arguments are definitely much more modern than what was produced by the Experimenters of the *Accademia del Cimento*. His language is easy to understand for us today, and it is worth quoting some parts that demonstrate the evolution of basic imaginative structures of our understanding of thermal phenomena.

Let us begin with Black’s clear criticism of the notion of *Cold*: “[…] let us examine what we mean by this quality of coldness. We mean a quality, or condition, by which the ice produces a disagreeable sensation in the hand which touches it; to which sensation we give the name of cold, and consider it as contrary to heat, and to be as much a reality. So far, we are right. The sensation of cold in our organs is no doubt as real a feeling as the sensation of heat. But if we thence conclude that it must be produced by an active or positive cause, an emanation from the ice into our organs, or in any other way than by a diminution of heat, we form a hasty judgment” [105]. He continues on p. 28 by saying that “[w]e are therefore under the necessity of concluding from these facts, that our sensations of heat and cold do not depend on two different active causes, or positive qualities, in those bodies which excite these sensations, but upon certain differences of heat between those bodies and our organs”. Note that, in the last part of the sentence, Black uses the word *heat* when he obviously means hotness—even though his work demonstrates that he knows the difference. He is just as careless as we are today when we believe that precision of expression does not really matter.

Here is this famous example where he shows how to distinguish between temperature and amounts of heat (p. 78): “[i]f, for example, we have one pound of water in one vessel, and two pound in another, and these two quantities of water are equally hot, as examined by the thermometer, it is evident, that the two pounds must contain twice the quantity of heat that is contained in one pound”. Moreover, with regard to the idea of heat capacity, his words betray a perfectly clear case of what became known as the Caloric Theory of Heat (p. 82): “[t]he quicksilver, therefore, may be said to have less capacity for the matter of heat”.

Joseph Black seems to be the one who coined the term *latent heat* (p. 127). Black J., 1803, vol. I, p. 127. He only applied it to phase change, particularly freezing and melting. Here is an excerpt from his writings that nicely shows how we can use natural language in order to speak about these phenomena (p. 129,130): “*[t]his experiment shews, that when water is cooled in a state of perfect rest, in a small vessel, it is disposed to retain this concealed heat, which I have been used to call its latent heat, a little more strongly than in ordinary circumstances. In common circumstances, the water retains the whole of this heat, until it be cooled to the 32d degree of Fahrenheit, or a very little lower. If, in ordinary circumstances, we attempt to make it colder, we may perhaps succeed in making it still colder by one degree or two, but no more, for then the latent heat begins to be extricated from a small part of the water, and to appear in the form of sensible heat, that small portion of the water which loses it, assuming consequently the form of ice.*” 

What Black describes here is the sub-cooling of water (Black spoke of over-cooled water). He used the term *latent heat* for *heat* stored in a material without making itself felt through its thermal intensity (i.e., its temperature). When *heat* is “felt” in the ordinary sense, it is called *sensible*. Researchers from Black onward, until well into the 19th century, talked about converting *latent heat* into *sensible heat* and vice-versa. Let us be clear that there are no different “forms” of *heat*—there is only the one extensive thermal quantity we call *heat*, which can simply have different effects.

### 3.3. The EQH of Simple Gases: From Laplace to Carnot

We now describe the notion of *quantity of heat* as it was used by Biot, Laplace, and Poisson (in the context of the theory of the speed of sound), and by Carnot as part of his theory of heat engines. Readers will be aware that the concept of quantity of heat used at that time was that contained in the caloric theory [107].

Before we discuss the aspects of interest to us, though, it makes sense to briefly describe what we should expect from writings in the period from about 1780 to 1825. This will allow us to interpret more easily what the qualitative statements of the researchers concerning an *EQH* might mean. Spelling out the descriptions and formalisms of the investigators, and understanding what they all have in common, is not always an easy matter.

#### 3.3.1. The Common Heritage

It turns out that, before a theory of thermodynamics—a theory relating heat to its power—was attempted by Carnot, all the scientists of interest to us shared a small number of assumptions regarding the thermal behavior of simple gases: (1) CTH: there exists a fluidlike quantity (called *quantity of heat* or *caloric*) that is responsible for thermal phenomena [108], (2) there exists a relation between pressure, volume, and temperature of such simple gases, and (3) changes of the caloric in a body of gas are related to changes of volume and temperature of the fluid body by two constitutive quantities called *latent heat* and *heat capacity* (what we are going to call the *LHHC rule*).

(1) In the CTH of gases, it was always assumed that *quantity of heat C*(*V,T*) of a body of gas was a “subtle” fluid that could be stored in the body, flow in and out, and would be conserved in all processes; therefore, from Equation (1), we conclude that for the calorists:(2)$$\dot{C}=\sum I_{C,i}$$

The term *heating* means that *I_C_* 0; *cooling* means *I_C_* 0. Since the caloric is conserved, there is no production term in the equation of balance. Moreover, we have left the source term out to reflect that the researchers at that time talked of conduction and radiation simply as two forms of transfer of caloric. 

(2) Second, gases were assumed to obey an equation of state. For an *ideal gas*, a modern form of the equation of state of is:(3)pV=nRT

Here, *p*, *V*, *T* are pressure, volume, and (absolute) temperature of a body of this gas [109,110,111,112]; *n* and *R* are the amount of substance of the body of gas in question and the universal gas constant, respectively.

(3) Finally, there is the *LHHC rule* [113,114] as the basic rule of calorimetry. J. Ivory ([115], p. 90) gives us a vivid description of the meaning of *latent* and *heat capacity* in the context of gases: “[…] the absolute heat which causes a given rise of temperature, or a given dilatation, is resolvable into two distinct parts; of which one is capable of producing the given rise of temperature, when the volume of the air remains constant; and the other enters into the air, and somehow unites with it while it is expanding […] The first may be called the *heat of temperature*; and the second might very properly be named the *heat of expansion*; but I shall use the well-known term, *latent heat*, understanding by it the heat that accumulates in a mass of air when the volume increases, and is again extricated from it when the volume decreases” (emphases in the original). In the modern form, it states that
(4)$$\dot{C}=\Lambda_{V}\dot{V}+K_{V}\dot{T}$$ (see [2], Section 5.2.2). The factor $\Lambda_{V}$ is called the *latent heat with respect to volume*, and *K_V_* is the *heat capacity at constant volume* [116]. This is a direct formal translation of J. Ivory’s [115] narrative description of the meaning of latent and specific heats. With this common heritage acknowledged, we can attempt to understand how Biot, Laplace, and others worked with the concept of the *EQH*, which they all called, indifferently, (*quantité de*) *chaleur* (heat, quantity of heat) or *calorique* (caloric; [117,118]).

#### 3.3.2. Lavoisier and Laplace

The investigators that followed the mid 18th century developments transferred the notion of *latent heat* to the expansion and compression of simple gases at a constant temperature—a subject that should prove important for thermodynamics. M. Lavoisier and P. S. Laplace, in a joint paper of 1783 [119], may have been the first to clearly state that a quantity of heat was absorbed in an expanding gas—and not just in fusion and vaporization—without making itself known to the thermometer (p. 388): “[s]ince dilation, fusion and vaporization are all effects of heat, it can be presumed with great plausibility that in the production of the first of these effects, as in that of the other two, there is a quantity of heat which is absorbed, and which consequently ceases to be sensitive to the thermometer…” [120]

In the case of (ideal) compressible fluids, the concepts of latent and specific heats (heat capacity) led to a simple image of the effect of a quantity of heat upon volume and temperature of gases. In 1816, Laplace [121] applied this to a mathematical theory of adiabatic processes of ideal fluids, and so derived our current model of the speed of sound in such media (as a correction to Newton’s formula which would apply if soundwaves were isothermal).

#### 3.3.3. Biot, Laplace, and Poisson on the Speed of Sound

Jean Baptiste Biot (1802, [122], p. 173) was apparently urged by Laplace to undertake an investigation into the question of why the speed of sound measured was higher than that predicted by Newton’s theory (based on the assumption that whatever happens to the air when sound passes through it would leave its temperature unchanged). Here is what Biot ([122], p. 176) had to say about heat, gases, and temperature: “[i]t is a fact known to physicists that atmospheric air loses part of its latent heat when it is condensed, and that on the contrary, when it is rarefied, it takes back a portion of sensible heat that it converts into latent heat” [123].

In P. S. Laplace’s “*Sur la Vitesse du Son*…” [121], we read on p. 238: “[w]hen its temperature is raised, while its pressure remains the same, only part of the caloric it receives is used to produce this effect: the other part, which becomes latent, serves to expand its volume” [124]. Laplace obviously makes use of the idea of latent heat of a gas, in accordance with the *LHHC rule*, Equation (4). 

He goes on (p. 239) to argue that in sonic vibrations, “[…] la quantité de chaleur reste la même […]” (“[…] the quantity of heat remains the same […]”). He then quickly presents his theorem for the speed of sound (p. 239), namely, that it is greater than the result derived by Newton by a factor equaling the square root of the ratio of the heat capacities at constant pressure and at constant volume. We shall present in further detail a brief outline of what must have led to this conclusion below.

In 1808, and again in 1823, M. Poisson worked on the theory of sound. Moreover, in 1823, he published another paper with the title “*Sur la chaleur des gaz et des vapeurs*” [125]. There, on page 340, we read: “[t]hese equations (5) contain the laws of elasticity and temperature of gases, compressed or expanded without variation in their quantity of heat; which will take place when the gases are contained in vessels impermeable to heat, or, when the compression, as in the phenomenon of sound, will be so rapid that one can suppose the loss of heat to be substantially nil” [126].

Summing up these brief excerpts, we can say that these researchers explained adiabatic compression and expansion of air as leaving the quantity of heat (amount of caloric) in a body of gas constant. Compressing air “converts” to change some latent caloric into sensible caloric, thus making the temperature rise, expanding air lets some sensible caloric “change” into the latent caloric allowing the temperature drop (see Section 5 on how to use this simple and imagistic explanation for teaching about adiabatic processes in introductory thermodynamics).

#### 3.3.4. Adiabatic Processes and the Speed of Sound

We have quoted Laplace’s ([121], p. 239) result that the actual speed of sound should be greater by the square of the ratio of the heat capacities (at constant pressure and at constant volume) than the value derived by Newton (*Principia*, 1687; [127], Liber Secundus, Sectio VIII, De motu per fluida propagato, pp. 357–374) for the propagation of pulses in an elastic medium that was based upon the assumption that the temperature remained constant, and the gas was ideal (the numerical result would be about 290 m/s instead of 340 m/s at 20 °C).

Laplace was cryptic about derivations and clouded everything in special assumptions concerning the material and mechanical nature of caloric; however, we might assume that he derived the relation between volume and pressure in adiabatic change by considering it to be composed of two steps (which, for easier derivation, we can visualize in a *p-V* diagram and/or a *T-V* diagram; [128]): (1) compression and removal of caloric (cooling) at constant pressure, and (2) addition of caloric (heating) at constant volume, so that in the end, the change in caloric would be equal to zero (i.e., $∆C_{1}+∆C_{2}=0)$. On the assumption of the existence of caloric, finite steps would lead to $∆C_{1}=K_{p}∆T_{1}$ and $∆C_{2}=K_{V}∆T_{2}$. The total change in temperature ∆T would be the sum $∆T_{1}+∆T_{2}$. Combining all the foregoing assumptions and relations, we obtain $∆T=∆T_{1}-\left(K_{p}/K_{V}\right)∆T_{1}$. Next, we apply the equation of a state of an ideal gas as in Equation (3) for calculating ∆T and $∆T_{1}$ in terms of the changes to ∆V and ∆p; we obtain nR∆T=p∆V+V∆p and $nR∆T_{1}=p∆V$, respectively. Finally, we have
(5)VΔp+γpΔV=0
where $\gamma =K_{p}/K_{V}$ is the adiabatic exponent. If this is written as a differential equation and integrated (and assuming that γ is constant), we get the well-known relations between volume, pressure, and temperature for adiabatic changes in a body of ideal gas [128]. A modern straight-forward derivation using the constitutive quantities of the ideal gas and the adiabatic relation(s) based upon Equations (2)–(4) is found in [2], Section 5.2.

What is important for later thermodynamics is this: the result can be derived from Equations (2)–(4) (and the equivalent version of (4) with pressure and temperature as an independent variable) alone. No relation between heat and the power of heat, and certainly no energy principle—which, at any rate, did not exist at that time—is needed. Moreover, the result provides us with information concerning the four constitutive quantities of a gas (caloric capacities at constant volume and pressure, and latent heats with respect to volume and to pressure). Since only two of the four quantities are independent, and the other two follow from these, this result reduces the task of finding—by measurement or by theory, as we shall see—the missing information.

#### 3.3.5. Fourier and the Conduction of Heat

In 1822, Jean-Baptiste-Joseph Fourier published his book titled *Théorie analytique de la chaleur* [129]. On pp. iii–iv, he writes: “I have concluded that in order to determine in number the most varied movements of heat, it is sufficient to subject each substance to three fundamental observations. Indeed, the different bodies do not possess in the same degree the faculty of containing heat, of receiving it, or of transmitting it through their surface, and of conducting it into the interior of the mass. These are three specific qualities that our theory clearly distinguishes, and which it teaches to measure” [130].

Even more clearly, directly, and schematically than anything we have heard before, Fourier states his assumptions concerning the schematic properties of *chaleur* (caloric): it is *contained* in bodies, the bodies can *receive* it (*transmit* it across their surfaces), and they can *conduct* it in their interiors. From these we will build our schematic principle of caloric as an *EQH*, which is useful for a dynamical theory of heat (see Section 3.5).

On the assumptions that (1) heat (caloric) is conserved when it is conducted, (2) a conductive current is given by $I_{C}=\lambda_{C}\Delta T$, and (3) both the heat capacity K (the caloric capacity!) and the caloric conductivity $\lambda_{C}$ of the material are constant, we obtain the well-known partial differential equation for the temperature field as a function of position and time; in single-dimensional conduction, we have: (6)$$\rho \kappa \frac{\partial T}{\partial t}=\frac{\partial }{\partial x}\left(\lambda_{C}\frac{\partial T}{\partial x}\right)$$

Here, ρ and κ are the mass density and the caloric capacity per unit mass, respectively. As we know, and as we shall discuss further below in Section 3.5, heat (caloric) cannot be conserved in conduction—conduction of heat is one of the important dissipative processes. Interestingly, following from Carnot’s thermodynamics, if we accept the result for caloric capacity and conductance (see Section 3.4.4), then a material having a constant temperature coefficient of energy and constant “energy” conductance has a caloric capacity and a caloric conductivity that are inversely proportional to temperature; in that case, dissipation exactly cancels the temperature dependence of κ and $\lambda_{C}$ in such a way that we obtain the same result as in Equation (6), and once again [2], Chapter 13.

Akin to the results concerning adiabatic changes and the speed of sound obtained by Laplace, Fourier’s theory of the conduction of heat does not depend upon a theory of thermodynamics and what we take heat to be, whether it is caloric or a quantity relating to energy. For our present purpose, however, the important point is the power and utility of the imaginative approach to quantity of heat as the *EQH* afforded by the caloric theory; this was a part of all the work we have considered here.

#### 3.3.6. Carnot and Caloric

Before we discuss Carnot’s important contribution to the science of *Heat*—the first known attempt at a theory of *thermodynamics* where the issue of the *Power of Heat* is clarified—we shall briefly see how he handled the notion of caloric in the context of gases. The following excerpt shows how he used his image of caloric as the *EQH* in order to reason with the behavior of gases [25], pp. 29–32: “[i]f, when a gas has risen in temperature by the effect of compression, we want to bring it back to its original temperature without making its volume undergo further changes, we must remove some caloric. This caloric could also be removed as the compression is carried out, so that the temperature of the gas remains constant. Similarly, if the gas is rarefied, it can be prevented from decreasing in temperature by providing a certain amount of caloric. We will call the caloric used in these occasions, where no change of temperature is made, caloric due to the change of volume. This denomination does not indicate that the caloric belongs to the volume; it does not belong to it any more than it belongs to the pressure and could just as well be called caloric due to the change of pressure” [131].

We can understand the meaning of this paragraph of Carnot’s in terms of Equations (2)–(4). First sentence: lowering the temperature at constant volume: dT/dt0 and dV/dt=0 (by (4)) → dC/dt0 (by (2)) → $I_{C}0$ (→ cooling). Second sentence: compressing and keeping the temperature constant: dV/dt0 and dT/dt=0 (by (4)) → dC/dt0 (by (2)) → $I_{C}0$ (→ cooling). Third sentence: isothermal expansion with heating (reverse of second sentence).

#### 3.3.7. Summary

It is quite clear now how the calorists, including Carnot, thought of and used the schematic (abstract) aspects of an *EQH* which they called *caloric* or (quantity of) *heat*. As we have emphasized, these aspects are simply those we can associate with a fluidlike quantity, nothing more, nothing less. In this generality, imagination will help us formulate a law of balance of the *EQH* as in Equation (1). This will help us create a theory of thermodynamics based upon a direct approach to entropy as the *EQH*.

### 3.4. Carnot: The Power of Heat

We have seen how, in 1824, Carnot [25] treated caloric in its schematic form. He made use of a simple and abstract version—one that employs just a few basic schematic elements—free of assumptions concerning details of materiality; all his results for a theory of thermodynamics can be obtained if we assume that much.

#### 3.4.1. Caloric Falling through a Temperature Difference

With Carnot, the story of a fluidlike quantity of heat continues beyond what his predecessors and contemporaries did: he used the imagery concerning the nature of heat in his analogy of waterfalls driving hydraulic engines and the workings of caloric in thermal engines. Indeed, he argued how the *Power of Heat* [132,133,134] could be understood imaginatively (and also formally) as *the power of caloric falling through a temperature difference* (Carnot, [25], p. 28): “[a]ccording to the notions established up to now, we can compare with some accuracy the driving power of heat to that of a waterfall: both have a maximum that cannot be exceeded, whatever the machine used to receive the action of the water, and whatever the substance used to receive the action of the heat. The driving power of a waterfall depends on its height and the quantity of liquid; the driving power of the heat also depends on the quantity of caloric employed and on what could be called, in effect, the height of its fall [Carnot’s Footnote 1 in [25], see below], that is to say, on the difference in temperature of the bodies between which the transfer of caloric takes place. In the waterfall, the driving power is strictly proportional to the difference in level between the upper and lower reservoirs. In the caloric fall, the motive power undoubtedly increases with the difference in temperature between the hot and cold bodies; but we do not know if it is proportional to this difference.

“(Carnot’s Footnote 1) The subject matter here treated being quite new, we are forced to use expressions which are rather unusual, and which perhaps do not have all the desired clarity” [135,136].

Before we take a look at some of the formalism afforded by Carnot’s imagery of heat falling from a higher to a lower level, we shall briefly review how accepting the existence of caloric (i.e., the *EQH*) shaped his reasoning. This is important if we wish to understand the steps taken in his analysis, which we summarize below.

#### 3.4.2. The Utility of Caloric as EQH

Carnot tells us that the *motive power of heat* does not result from “using up” caloric but only from passing it from a hot to a cold body ([25], pp. 10,11; emphasis in the original): “[t]he production of motive power is thus due, in steam engines, not to an actual consumption of caloric, *but to its transport from a hot body to a cold body* …” [137]

The transport will be effected by a fluid (such as an ideal gas) serving as the working agent in the heat engine; this gas will undergo the famous Carnot cycle (a description is given further below): “[w]e implicitly suppose, in our demonstration, that when a body has undergone any changes, and […] is brought back identically to its primitive state, […] we will suppose […] that this body is found to contain the same quantity of heat that it contained at first, or otherwise that the quantities of heat absorbed or released in its various transformations are exactly balanced” (Carnot, ([25], p. 37, footnote; [138]). This means that the agent receives some caloric from the hot body (the furnace), briefly keeps it, and then emits exactly the same amount to the cold body (the cooler). After a single complete cyclic operation, the fluid body contains as much caloric as it did initially. Put more formally, there is a heat function $C\left(V,T\right)$ [139] whose difference on a segment of the cycle equals the net amount of caloric transferred during that part of the process. Additionally, for a complete cycle, $C_{abs}=-C_{em}$ (abs: absorbed; em: emitted).

Moreover, the power is independent of the agent used in the heat engine ([25], p. 38): “[t]he motive power of heat is independent of the agents used to realize it; its quantity is determined solely by the temperatures of the bodies between which the caloric is ultimately transported”. [140]

In sum, the *Power of Heat* is determined by the quantity of caloric falling from a body at a high temperature to a body at a low temperature; it is proportional to the amount of heat (caloric) falling through the temperature difference, and it depends upon this drop in temperature, but we do not know how exactly. To the extent that the assumptions contained in his words are meant to be formal, Carnot uses the following relation for the *Power of Heat* [141]:(7)$$P_{th}=\left[F\left(T_{high}\right)-F\left(T_{low}\right)\right]I_{C}$$

We call $F\left(T\right)$ *Carnot’s function* which is a universal function of temperature (i.e., it is independent of the fluid used). Carnot’s program to determine the concrete form of $F\left(T\right)$ consists of an analysis of the action of caloric in a heat engine facilitated by a working fluid (he always uses the ideal gas) going through the Carnot cycle. Interestingly, apart from finding Carnot’s function F, the theoretical work will fix the form of the latent heat with respect to volume ($\lambda_{V}$) and the other three caloric constitutive quantities of the ideal gas as well [142].

#### 3.4.3. Carnot’s Cycle and Uniform Gaseous Bodies

The Carnot cycle, which is undergone by a body of a simple gas, is well known; nevertheless, there is an issue worth discussing—it mostly relates to questions having to do with dissipation, and how Carnot, with his imagery and his cycle, avoided having to deal with it directly. Basically, the problem is avoided if we never bring bodies at different temperatures in direct contact with one another, and if we can consider the working fluid a uniform body at all times.

To be clear, a body of (ideal) gas undergoing a Carnot cycle first absorbs caloric at a constant high temperature (and therefore it expands); second, it expands further without heating or cooling, which lets its temperature drop to its low value (adiabatic expansion); third, the body rejects the caloric absorbed during step 1, as it is compressed at constant low temperature. Finally, adiabatic compression brings it back to its initial high temperature. What is important here is the absorption and rejection of caloric taking place at *constant* temperatures.

Carnot was motivated by his image of “falling” caloric, which he expressed as follows ([25], p. 9): “[t]he production of motion in steam engines is always accompanied by a circumstance […] which is the re-establishment of equilibrium in the caloric, that is to say its passage from a body where the temperature is more or less high to another where it is lower” [143]. Moreover, whenever caloric is allowed to pass from a hotter to a colder body, whenever there is a temperature difference, motive power *may* be produced ([25], p. 12): “[w]herever there is a difference in temperature, wherever there can be a restoration of equilibrium in the caloric, there can also be production of motive power” [144]; however ([25], p. 23), “[s]ince any restoration of equilibrium in the caloric can be the cause of the production of motive power, any restoration of equilibrium without the production of this power must be considered as a real loss” [145]. In other words, motive power *need not* be produced! If it is not, if we let caloric pass from a hot to a cold body without motive power having been produced, we have a “true loss” of (motive) power [146].

If we assume, most likely with Carnot, that caloric comes from a hot body at a given fixed temperature, then the perfect way of transferring it to a lower temperature without incurring “a true loss” is for the working fluid to be heated at this constant upper temperature. Likewise, at the lower end, the caloric absorbed during the first step needs to be discharged at the constant temperature of the environment (or cooler).

Furthermore, there should not be any “re-establishment of equilibrium in the caloric without the production of motive power” when caloric is distributed inside the working fluid. In reality, it needs to flow from warmer to cooler places inside the fluid if it is to be evenly distributed. By accepting two assumptions, Carnot avoids this difficulty. First, in his entire theory, he obviously accepts the model of spatially uniform bodies: a body of gas has single values of temperature and pressure at any given moment. Second, to make such a model “believable,” he invented the image of materials that *lets caloric pass easily*. When Carnot describes his cycle, he says ([25], pp. 32,33): “1° Contact of the body A with the air enclosed in the capacity abcd, or with the wall of this capacity, a wall which we shall suppose to *transmit the caloric easily* (emphasis ours). The air is by this contact at the same temperature of the body A; cd is the actual position of the piston. 2° The piston rises gradually and takes the position ef. The contact always takes place between body A and the air, which is thus maintained at a constant temperature during the rarefaction. The body A provides the caloric necessary to maintain the constant temperature” [147]. This is (part of) Carnot’s reasoning for caloric being the *EQH*.

#### 3.4.4. Analysis of the Action of Caloric in Carnot’s Cycle

The analysis roughly proceeds as follows. Carnot applied Equation (7) to a cycle with short adiabats (i.e., for small ∆T); in this case, the *energy made available* in one cycle is given by $F^{\prime }C_{abs}∆T$ ($F^{\prime }$ is the derivative of F with respect to T); see Section 4.1 for a discussion of energy made available. If we use an ideal gas, this is also equal to $nR\ln\left(V_{2}/V_{1}\right)∆T$ (where $V_{2}$ and $V_{1}$ are the end and starting points of the volume of gas, respectively, in the isothermal expansion representing the first step of the Carnot cycle). This tells us that $C_{abs}=nR/F^{\prime }\ln\left(V_{2}/V_{1}\right)$. Since $C_{abs}$ is also determined by the integral of the latent heat (caloric) over the volume (as the volume goes from $V_{1}$ to $V_{2}$), we have
(8)$$\Lambda_{V}=\frac{1}{F^{\prime }}\frac{nR}{V}$$

In sum, we find $F\left(T\right)$ if we first accept that $C_{abs}$ is equal to the difference of the heat function at the end and the beginning of the step of isothermal expansion. This allows us to conclude that the caloric capacity at constant volume ($K_{V}$), which is the derivative of the heat function with respect to temperature, must be equal to $K_{V}=nR\cdot {\left(1/F^{\prime }\right)}^{\prime }\ln\left(V/V_{1}\right)+C{}^{\prime }\left(V_{1},T\right)$ ($V_{1}$ is fixed; it represents the starting point for V of the Carnot cycle; again, ()’ denotes a derivative with respect to temperature). Moreover, from Equations (3) and (4) and their transformations, we obtain $K_{V}=a/\left(F^{\prime }T\right)$ (a= const) if we *assume that the adiabatic coefficient is constant*. In other words, $K_{V}$ is a function of T only. From the foregoing relation for $K_{V}$ (where it may also depend upon V) we conclude that $F^{\prime }=$ const and F=cT+d. If we set c=1, we can write the following expression for *thermal power* $P_{th}$:(9)$$P_{th}=\left(T_{high}-T_{low}\right)I_{C}$$
and for the latent caloric:(10)$$\Lambda_{V}=\frac{nR}{V}$$

In other words, the *power of a fall of caloric is proportional to the difference of the upper and lower temperatures*; this is what we should expect if we were to accept the analogy between the power of a waterfall and the power of a fall of caloric by projecting the role of a gravitational potential difference to a thermal tension (i.e., to a temperature difference).

This, plus knowledge of the ratio of the heat (caloric) capacities from measurements of adiabatic processes, fixes all four constitutive quantities of a simple gas. In particular, the caloric capacities are given by
(11)$$K_{V}=\frac{1}{\gamma -1}\frac{nR}{T}$$
and
(12)$$K_{P}=\frac{\gamma }{\gamma -1}\frac{nR}{T}$$

These results let us calculate all that is necessary if we wish to work out applications of the thermodynamics of simple gases. 

Carnot did obtain the results presented here [148] in Equations (9)–(12), but he seems to have had doubts as to their validity. We may assume that what caused this doubt, and what most likely killed the caloric theory (see Section 3.5), is the fact that in his thermodynamics, the specific heats (heat capacities: caloric capacities) of an ideal gas are not constant; indeed, they are inversely proportional to the absolute temperature, making them diverge for *T* → 0 K. In contrast to the calorists and their successors, we do not need to worry about this.

### 3.5. The Trouble with the Caloric Theory of Heat (CTH)

We have seen three important successes of a theory of heat based upon a direct approach to an *Extensive Quantity of Heat* (in the form of caloric): (1) results for adiabatic processes and the speed of sound, (2) a theory of the *Power of Heat*, and (3) a theory of pure conduction of heat. In the first two cases, the fluids involved have been assumed to be ideal—caloric is conserved in their operations. In the case of the conduction of heat, however, the “success” needs some extra explanation: conduction is clearly dissipative, and we must ask what this means for a caloric model of heat conduction. Moreover, we want to understand why the CTH was rejected by Carnot’s successors.

#### 3.5.1. Troubling Assumptions, a Solution, and a Rival to the CTH

If we are tempted to look to caloric as a model for the *EQH*, we first need to understand aspects of this concept—as it was conceived of in the period covered in our story—that could spell trouble for, limit, or even invalidate the idea of caloric. There are two very different types of assumptions that cause rather different types of difficulties: one results from a distracting philosophical (epistemological) fallacy, the other concerns a serious limitation of assumed fundamental schematic properties of caloric. The former has to do with special mechanistic and “atomistic” models of caloric [69,97,149]; the latter makes caloric a conserved quantity.

These assumptions are fatefully intertwined: it is very likely that calorists assumed caloric to be a conserved quantity because they took the naïve realistic stance of making caloric a mechanical “quasi-material” entity, possibly composed of “particles”. Since the assumptions concerning the “material nature” of caloric are unnecessary (as demonstrated already by Poisson and Carnot), we can rid ourselves of this form of naïve realism. This will free us to consider what is otherwise “unthinkable:” caloric does not need to be conserved.

In summary, two steps must be taken if we wish to make caloric a viable candidate for the *EQH*: we need to (1) switch from naïve realism to a form of epistemology allowing for the central role of imagination and figurative thought—which we might call embodied realism [11]—freeing us to accept caloric as a “metaphoric” fluidlike quantity; and (2) add production (but not destruction) to the list of schematic properties of the concept of caloric—which otherwise is composed of the schemas of storage and flow. We shall call the resulting model of caloric the *Extended Caloric Theory of Heat* (ECTH).

This sounds deceptively simple, which makes us wonder why, apart from appealing to the power of naïve realism, these steps were not taken in the period around and after 1825. There is another, maybe even more powerful reason for not taking these steps: our predilection with “little particles” and the desire to explain natural processes in terms of theories that are not just analogous to Newton’s mechanics—the most successful mathematical theory ever created up to that date—but actual versions of mechanics.

In short, the main reason for the course of events was a rival metaphor for *heat*, neatly summarized in Clausius’ statement of 1850 ([150], pp. 369,370) “[…] that in recent times more and more facts have become known which speak in favor of heat not being a substance but consisting in a movement of the smallest parts of the bodies” [151]. Having this statement in the paper that established the *mechanical theory of heat* in TET is very interesting considering that the mechanical theory is a *macroscopic* model of thermal phenomena. Clausius accepted Carnot’s very general idea that the power of a heat engine had to follow from the strength of the flow of heat coming from the furnace and the upper and lower temperatures involved [152]; however, he re-conceptualized the nature of heat. Heat was now supposed to be (partially) interconvertible with work (i.e., a form of energy appearing in heating and cooling); as a consequence, only part of the heat communicated to the engine from the furnace is rejected to the environment at the lower temperature—the rest has been “converted” into work.

The step of making heat (partially) interconvertible with work needed to be motivated: Clausius must have found his reason in the issue of the heat capacities of a simple gas. Remember that, in the caloric theory, heat capacities are not constant; see in Equations (11) and (12). On the other hand, a simple “vis viva” model of heat leads to constant *heat capacities* of a simple gas—exactly what was desired. Indeed, in the mechanical theory of heat, the heat capacities of a simple gas would turn out to be constant.

Considering that experimental evidence for either constant or variable heat capacities was lacking at the time (see below, Section 3.5.2), an assumption about the nature of heat was needed, and it was found in an embryonic microscopic model. What the model lacked in experiential basis, it more than made up for in “natural philosophical” appeal. Moreover, the real rival of the caloric theory of heat was never the macroscopic mechanical theory founded by Kelvin and Clausius, but what later became an important example of microscopic theories in physical science.

#### 3.5.2. Concrete Trouble with the Caloric Theory of Heat (CTH)

Remember that the only remaining practical challenge to the CTH is the question of whether caloric is conserved. The additional troubling assumption discussed above in Section 3.5.1, is dealt with if we forgo forms of naïve realism in favor of an explicitly imaginative approach to the *EQH*.

Thus, what kind of actual practical trouble are we getting into if we assume that caloric is conserved? We shall briefly discuss three points: the production of heat in mechanical friction, the temperature dependence of the heat capacities (the caloric capacities), and the practice of calorimetry where quantities of caloric are supposed to be measured by bringing bodies with different temperatures into thermal contact.

***Mechanical friction.*** Much has been made of Rumford’s cannon boring work that seemed to show that limitless amounts of caloric would develop if we just continued boring forever. What seems to be clear to us today is that a clear and damning observation was apparently brushed aside by all calorists in the period where their important work was done. The argument was simple: materials contained limitless amounts of latent heat that was “developed” in phenomena such as the cannon boring experiments and combustion. “Developing” meant that latent heat was “converted” into sensible heat.

Naturally, our modern explanation will be different: heat (quantity of heat, our *EQH*) is produced in irreversible processes such as mechanical friction. An *Extended Caloric Theory of Heat* (ECTH) can deliver the goods: a model of caloric basically equal to the abstract version used by Carnot, augmented by the assumption that caloric is produced in irreversible processes.

***The temperature dependence of the specific heats.*** As we have seen, in Carnot’s theory of the power of heat, the caloric theory leads to specific heats (caloric capacities) of the ideal gas that are inversely proportional to their (absolute) temperature; see Equations (11) and (12). We could have at least two different reactions to this. First, we do not like such a state of affairs; what if fluids obeying Equation (3) could exist all the way to absolute zero? Now, since this does not happen—gases liquefy long before this—we do not have to worry about this. 

Second, we could accept a completely different view of the nature of heat, namely, that *heat* needs to be understood as the vis viva of the motion of little particles that make up our materials. This is what happened at the hands of the later investigators. Accepting this, the simplest model of non-interacting particles roaming in empty space suggests that the specific “heats” (i.e., the temperature coefficients of energy and enthalpy), should be constant. The trouble with this argument—if it is directed against the caloric theory—is that we have completely left the perspective of an *EQH* (upon which the schematic and metaphoric structure rests which we have explored here) and constructed a wholly different metaphorical foundation. As we discussed at the end of Section 2.5, abandoning one perspective to construct a totally different one cannot serve as an argument against the former. All we could argue is that we *want* the specific heats to be constant, and that is why we *do not want* the caloric theory to work.

***Calorimetry: Measuring amount of caloric.*** However, there is a more serious point to be made. If we assume that the usual calorimetric measurements measure quantities of caloric, the caloric theory (in its non-extended version) is dead. If we, for example, say that the amount of caloric emitted by a warm body of water in thermal contact with a cold body of water is equal to the amount received by the colder body, we will conclude that the caloric capacity of water is (close to) being independent of temperature. If the researchers in the period we have discussed in this section had ever managed to measure the specific heats of air with sufficient accuracy using their calorimetric equipment, they would indeed have concluded that the (caloric) capacities had to be constant [153].

However, they would have been wrong if they had measured caloric capacities. If we accept the *Extended Caloric Theory* (ECTH), where caloric is produced in irreversible processes, we are safe. The calorimetric experiments are irreversible, and caloric will be produced. In the case of water, when the temperatures of two equal bodies of water equilibrate at the midpoint of hotness, the caloric capacities that have been determined will be smaller at higher temperatures because there is more caloric in the water than there was at the beginning.

## 4. The Science of Heat as a Force of Nature

We suggest that experiencing *Heat* as a *Force of Nature* makes available a concept of an *Extensive Quantity of Heat* (*EQH*); see Section 2. We further suggest that much of what we have seen developing in the historic period covered in Section 3 reflects this natural attitude: at least as far as the schematic (abstract) aspects of caloric are concerned, we can accept the historical version of caloric as the not yet extended version of an *EQH*. This is easily extended by including production in irreversible processes, leading to Equation (1). Motivated by what we have described, we shall now sketch an example of the thermodynamics of uniform dynamical systems that accepts an *EQH* as part of the basic elements from which the theory is built. By comparing the results with what can be found in continuum physics or the physics of uniform dynamical systems, we shall see that our *EQH* fits the entropy we know from modern approaches to thermal processes. We then ask if entropy is a resurrection of caloric in ECTH.

### 4.1. The Role of Energy in Physical Processes

Before we begin a very brief sketch of a theory of heat based directly upon an *EQH*, we need to discuss the role of energy in physical processes. It starts with, and then extends, the notion of power.

Readers may very well wonder why the concept of energy has not really made much of an appearance in our outline of the development of historical models of *Heat* in the period covered here (Section 3). It is true that power as a measure of the strength of the interactions between *Forces of Nature* (i.e., of *Heat* and *Motion*, as in Carnot’s theory) has been instrumental, but that alone does not yet present us with a complete energy principle. There are at least two reasons for our relative quiet on this front: first, a generalized version of a concept of energy did not exist and was obviously not needed for what was achieved before 1825; second, it should be clear that introducing models for the role of energy in our story would have distracted, at least at first, from our goal of unearthing reasons why we need, and how we can have access to, the notion of an *Extensive Quantity of Heat* [154,155,156].

The distraction would have resulted not just from opening up another line of (historical and conceptual) investigation but from going down the rabbit hole following a sentiment about how nature works that may very well be ancient but gained the strength, in the 1830s and 1840s, to overpower much of the work that had been done before that time. That sentiment is the source of the idea that *Forces*—Heat, Light, Electricity, Magnetism, Chemical Affinity, and Motion—are convertible into one another. Indeed, there might be but a single *Force*, and *Force* would eventually turn into the concept of *energy*; the world would, from then on, be divided into two entities: Matter and *energy*. This sentiment is very well expressed by W. R. Grove in a course of lectures, published with the title “*On the Correlation of Physical Forces*” in 1846 [155]. On p. 8, Grove writes: “The position which I seek to establish in this Essay is, that the various imponderable agencies, or the affections of matter which constitute the main objects of experimental physics, viz., Heat, Light, Electricity, Magnetism, Chemical Affinity, and Motion are all Correlative, or have a reciprocal dependence. That neither, taken abstractedly, can be said to be the essential or proximate cause of the others, but that either may, as a force, produce or *be convertible into the other* […].

“The term Force, although used in very different senses by different authors, in its limited sense may be defined as that which produces or resists Motion. Although strongly inclined to believe that the five other affections of matter, which I have above named, are, and will ultimately be resolved into, modes of motion, it would be going too far, at present, to assume their identity with it; I therefore use *the term Force, in reference to them, as meaning that active principle inseparable from matter, which induces its various changes*”. (Emphases added.).

There is a second “feeling” expressed in these lines: That, in the end, there will be but a single *Force*, namely *Motion*; everything else will “ultimately be resolved into, modes of motion”. Here, and in the notion of interconvertibility, we have, laid out clearly before our eyes, the program that became TET and much of modern popular science.

We may very well credit the notion of energy, as it was harbored by the scientists that came after Carnot (see T. S. Kuhn [156] for an early history of the energy principle), and as it made its way into physics in general and thermodynamics in particular, with laying a heavy veil over our basic image of how *Forces of Nature* work and interact. This veil effectively hid (and still hides) the extensive (fluidlike) quantities of physical science from view—so much so that in the science of heat it is mostly unknown. The popular science created around energy, taught in school and presented to us every single day in the media is more than just a small distraction. The banishment from view of a simple measure of an *EQH*, and a dimming of a clear understanding of electric charge and momentum in electric and mechanical processes, respectively, amply prove this point. What we get when we constantly mix up electricity and motion with energy—at least in every-day discourse—should be a warning sign on our path to concepts of thermal phenomena [39]

Nevertheless, it is clear that we need to have an understanding of the role of energy in physical processes—intensive and extensive quantities characterizing a *Force of Nature* will not go all the way ([2], Chapter 2). A discussion of the simple example of an electrically driven water pump shows what we mean (Figure 2). The electric quantities of interest—voltage and the strength of the electric current—are unrelated to the hydraulic ones—pressure difference and volume current. If we want to describe how electricity couples to water, we need a fifth quantity, power: assuming steady-state operation, the product of voltage and the electric current will equal to the product of pressure difference and volume current if the coupling is ideal. If it is not, the latter product will be less than the former. Although non-ideal coupling gives the impression that a sixth independent quantity (i.e., efficiency) is needed for a full description, this is not the case. Models of electric circuits and fluid flow in concrete settings involving concrete constitutive (material) relations will allow us to quantify dissipation. The statement remains correct: in addition to models of forces of nature involving their characteristics and associated constitutive quantities, we need the concept of power.

So far, we simply apply Carnot’s reasoning concerning processes, their coupling, and power. Now we need to make the decisive step: we interpret power (i.e., the strength of coupling of forces of nature) as the rate at which *energy* is either *made available* (i.e., by the driving processes, such as electricity) or *used* (i.e., by the caused processes, such as fluid flow and the production of entropy as a consequence of both electricity and fluid flow). Expressed differently, energy made available or used is the integral over time of the power of a process (either causing or caused). 

Now, if we imagine this energy as “something” handed from the causing agent to the caused patient, we may ask where the energy is coming from. This is where the concept of energy gets its important extension: it is supplied to the pump where the coupling of forces takes place. If we wonder what happens with energy, we find that it was exchanged in the coupling: it will be carried away from the pump. This “carrying” is done by the forces at play in a real pump (i.e., *Electricity*, *Fluid*, and *Heat*) in ways that can be quantified by constitutive relations specifying energy transfer in terms of the transfers of the extensive and intensive quantities related to the forces. So far, if we assume conservation of quantities of energy, the sum of energy currents with respect to the pump will be equal to zero for the steady-state operation.

This leaves a final step to be performed: in transient phases, inflows and outflows of energy are unbalanced. The assumption of energy conservation is saved by assuming that energy is either added to or withdrawn from storage—energy can be stored in physical systems; therefore, we can write a law of balance for energy in the form
(13)$$\dot{E}=I_{E,net}+\Sigma_{E}$$

The first term on the right is the sum of all conductive and convective energy currents, whereas the second term, a source rate, denotes the rate of energy transfer due to radiative interactions. This relation constitutes the element that goes beyond what Laplace, his contemporaries, and even Carnot had available. Except for a notion of power, there was no generalized notion of energy, nor of energy currents (and energy transferred), nor of an energy function (energy stored such as in a gaseous body; this was Clausius’s first important contribution to thermodynamics [150]).

### 4.2. Assumptions for a Model of Thermo-Fluid Processes

We now formulate the assumptions needed for creating a model of thermo-fluid processes undergone by a spatially uniform body of a viscous compressible fluid ([2], Section 10.1.2). As we have done from the start of this paper, see Equation (1) and [43], we shall write *C* for the amount of heat in the sense of our *EQH*, and assume it to be a function of T, V, and dV/dt. The dependence upon the rate of change of volume is needed, at least initially, since we want to discuss the thermodynamics of a viscous fluid: viscosity will make itself known as a consequence of the speed at which we change the volume of the fluid ([2], p. 458). We shall accept, just as Carnot did, that C spreads easily inside the fluid—if we did not do this, we could not be able to build a theory of spatially uniform bodies. 

Changes of volume, however, will be the source of dissipation (i.e., of production of C), if the fluid is viscous. In the following, $I_{C}$ will denote a current of C across the surface of the fluid body under consideration; $\Pi_{C}$ shall denote the production rate of C. In other words, we shall accept Equation (1), with the source term dropped, and no convective currents admitted.

Second, we accept a simple form of the balance of momentum: the rate of change of momentum is equal to the sum of the conductive momentum currents (i.e., the surface forces acting upon the fluid body as its volume changes (excluding body forces; here, *force* stands for the usual meaning of the word in mechanics ([2], Chapter 3)).

Third, we take for granted a general form of the law concerning the balance of energy for such fluid bodies where the rate of change of the energy is given by the sum of energy currents that result from (1) heating/cooling and (2) compression/expansion:(14)$$\dot{E}=I_{E,th}+I_{E,comp}$$

This is a special case of Equation (13). In our example, the energy of the viscous fluid is a function of T, V, and dV/dt, as are C and, in particular, the pressure.

This brings us to the fourth point: we shall assume a function of “state” of the fluid body that includes a viscous pressure term:(15)$$p\left(T,V,\dot{V}\right)=p{|}_{E}\left(T,V\right)+a\dot{V}$$

Here, $p{|}_{E}\left(T,V\right)$ is the “static” pressure of the fluid (i.e., the value of the pressure attained when the volume does not change or if viscosity is neglected ([2], Section 10.1.1)).

Fifth, we assume that the current of C is continuous across an ideal wall between two fluids (so that T is continuous as well). Assuming the existence of such walls is necessary for the measurement of temperature to work: one of the fluids separated by the wall would be the fluid of the thermometer, the other would be the body whose temperature we wish to measure (for the reasoning behind this assumption, see [27], pp. 168–169).

Sixth, we accept the usual expression for the energy current associated with the compression and expansion of a fluid body (the energy current is equal to the negative product of the pressure of the body and the rate of change of volume). Seventh, and quite importantly, we assume that a conductive energy current associated with heating or cooling is proportional to *I_C_* (this is a consequence of Carnot’s result for the power of *Heat*; [2], Section 4.4.2); however, we do not need to assume the form of the factor of proportionality, (i.e., of Carnot’s function) as this will be derived from our model.

### 4.3. Results of a Thermodynamics of Viscous Fluids

We do not go through the lengthy details of the derivation which makes use of the method of Lagrange multipliers ([2], Section 10.1.3), [157,158]; we simply summarize the results. First of all, the factor of proportionality between *I_E,th_* and *I_C_* is determined from general relations obtained in the derivation if we specialize them to the ideal gas as in Equation (3): the Lagrange multiplier for entropy turns out to be the absolute temperature ([2], Section 10.1.4); therefore, we recover Carnot’s determination of his function: *F*(*T*) = *T*, and Equation (9) for the power of a fall of a quantity of *C*. Hence,
(16)$$I_{E,th}=TI_{C}$$

Once this is obtained, a list of the most important additional results can be derived ([2], Section 10.1.5):(17)$$\begin{array}{c} \Pi_{C}=-\frac{a{\dot{V}}^{2}}{T},\;a\leq 0\\ I_{C}=\Lambda_{V}\dot{V}+K_{V}\dot{T}+\frac{a{\dot{V}}^{2}}{T}\\ \dot{E}=T\dot{C}-p{|}_{E}\dot{V} \end{array}$$

For a=0, we obtain the relations for ideal (non-dissipative) fluids. Moreover, for the ideal gas, the latent heat and heat capacity (always referring to our *EQH*) are given by Equations (10) and (11), as in Carnot’s analysis.

### 4.4. Entropy as the EQH, and the Question of Caloric

Readers accustomed to TET in the form of the *Gibbs Fundamental Form* which is applicable to the thermodynamics of simple fluids (i.e., *dE* = *TdS* – *pdV* [35,49]), will have noticed that our *EQH*, which we have abbreviated by *C*, is the *entropy* of Clausius’s and Gibbs’s thermodynamics. In actuality, however, it is much more than that: using our *EQH* as entropy, we can formulate models and theories that go beyond TET, such as a full-fledged thermodynamics of *Uniform Dynamical Systems* [2], *Continuum Thermodynamics* [26,27,28], and applications to *irreversible thermodynamics* such as those found in thermoelectricity [52,159,160,161] (see also Section 5).

#### 4.4.1. Energy and Entropy in Heating and in Dissipative Processes

A conductive current of energy “carried” by a conductive current of entropy equals the product of temperature and entropy currents:(18)$$I_{E}=TI_{S}$$

This is what we read from Equation (16). By the same token, the relation between dissipation rate (i.e., the rate at which energy is used in the production of entropy) and the entropy production rate is given by
(19)$$P_{diss}=T\Pi_{S}$$

Equation (17), first line, is a particular result for the more general expression presented here. Thus, if a home is losing energy at a rate of 600 W in winter, it loses entropy at a rate roughly equal to 2 W/K, and an electric immersion heater working at 300 W in water at a temperature of 300 K produces entropy at a rate of 1 W/K.

#### 4.4.2. Constitutive Quantities in Entropy and Energy Representations

Having developed an entropy representation of thermal processes, we will want to work with entropic constitutive (or material) quantities such as entropy capacity [2,162], latent entropy, entropy conductivity, and entropy transfer coefficients. This is what we have done above: we have introduced entropy capacitances $K_{V}$ and $K_{p}$, and latent entropies $\Lambda_{V}$ and $\Lambda_{p}$. We should do the same for entropy transfer: in models involving flow and transport of entropy, we introduce an entropy conductivity for conductive flow, and an entropy transfer coefficient for the typical cases of convection (and radiation) at solid-fluid boundaries.

Thermodynamic relations tell us that the following relation holds between entropic and energetic constitutive quantities (*CQ*):(20)$$CQ_{energy}=T\cdot CQ_{entropy}$$

Therefore, for a substance such as water where the energy capacity is pretty much constant, the entropy capacity varies inversely with temperature. At ambient pressures, the specific entropy capacity of water varies between 15.4 J/(K^2^kg) and 11.3 J/(K^2^kg) for temperatures between 0 °C and 100 °C, respectively. For a substance such as glycol where the specific entropy capacity is constant (about 7 J/(K^2^kg)), the energy capacity (the temperature coefficient of energy) rises in proportion to temperature. 

For thermal conductivity, we present the example of manganin [163] whose entropy conductivity is nearly constant and equal to about 0.068 W/(K^2^m) between 400 K and 600 K; this makes the energy based thermal conductivity rise from about 27 W/(K·m) to 40 W/(K·m). Finally, with a typical thermal energy transfer coefficient between a wall and the air in an apartment of 12 W/(K·m^2^), the corresponding entropy transfer coefficient equals 0.040 W/(K^2^m^2^).

There are some examples of material coefficients that demonstrate that it makes eminent sense to introduce them in terms of entropy representation. Take the case of thermoelectricity (which we will discuss in some more detail in Section 5), where we should expect reciprocity between generator and heat pumping operations. The former is described with the help of the Seebeck coefficient, the latter uses the Peltier coefficient. The two will be seen to be identical only if we define the Peltier coefficient *a* with respect to entropy (i.e., $I_{S}=\alpha I_{Q}$ [52]); in the energy representation, the relation will depend upon temperature.

A second, even more interesting, example showing the superiority of the entropy representation over its energy equivalent is the quantum of thermal conductance [164]. The entropy conductance is subject to a universal quantum, $\pi k_{B}/6\hbar$, whereas the value relating to energy conductance depends upon temperature, $\pi k_{B}T/6\hbar$, ($k_{B}$ is Boltzmann’s constant, and h Plank’s constant).

#### 4.4.3. Resurrecting Caloric?

Does this mean we should resurrect CTH naturally in the form of ECTH? After all, what we have seen happening during the period recounted in Section 3 demonstrates that the researchers based their reasoning upon elements of a “naturalistic” approach to experiencing *Heat* as a *Force of Nature*. If we abstract from their rather special assumptions concerning material and mechanical properties of caloric—which varied from researcher to researcher—we are left with what the schematizing action of our mind presents us with (Section 2). 

In the period since the consolidation of TET, several authors have pointed out similarities between Carnot’s caloric and Clausius’s entropy ([165,166,167,168]; but see [169,170,171]). Readers so inclined would find enough evidence for the utility of turning to a modernized and extended caloric theory (ECTH, Section 3.5)—we get there if we allow ourselves to be guided by the figurative approach supported by a schematizing mind (remember the description, in Section 2, of experiencing and conceptualizing). No matter, however, what our preferences with regard to caloric might be, we can certainly state that entropy can be understood as the *EQH* of thermal phenomena, which allows us to construct an *Experientially Natural* form of *Thermodynamics* (ENT).

## 5. Applications of a Direct Approach to Entropy as Quantity of Heat

An ENT should lead to models of thermal systems and processes much more directly than what we get from TET. In particular, since a science of *Forces of Nature* presents phenomena to us as dynamical, we can expect thermal processes to be expressed as initial value problems.

We shall briefly discuss three phenomena—adiabatic change, heat engines, and thermoelectricity—both from their conceptual and didactic aspects, and ask, as a form of summary, if a direct approach to entropy as the *EQH* is feasible at different educational levels and for different audiences. Moreover, in sub-Section 5.4, we demonstrate how entropy can be measured (made concrete in a high school lab activity).

### 5.1. Understanding and Modeling Adiabatic Change Undergone by Air

There is a general failure to (qualitatively) understand adiabatic processes in typical educational settings, including introductory university physics [172]. Since thermal processes are assumed to be the result of transfers between *heat* and *work*, and understood as (forms of) *energy*, we might be inclined to find remedies to this dismal state of learning [173] by strengthening the understanding of the role of work (i.e., by teaching mechanics more effectively [174]).

This completely misunderstands and misrepresents the role of basic abstractions and concrete knowledge learners make use of when interpreting experience. The fact that the temperature of air goes up when the fluid is quickly compressed, is correctly interpreted as having something to do with heat [175]. When asked for reasons for this phenomenon, laypersons and learners of physical and engineering sciences alike offer the idea that heat is produced by “the little particles of air rubbing against each other”. An engineering student put it like this ([175], p. 9): “[h]eat is generated by friction. I used to explain the phenomenon [of adiabatic compression] like this because I knew that heat is produced in friction”. After all, we are quick to understand that (1) heat is needed for raising the temperature of a material, and (2) heat can come from friction. Finally, since we are all modern humans, we (3) “know” that air is composed of molecules that move (and most likely “rub”).

However, when given the opportunity to reason with the help of a notion of an *EQH* called *heat*, a very different picture emerges. Here is a brief exchange between a physicist (P) and a layperson (L) ([175], p. 2): “P: If you compress air quickly what happens to its temperature? L: It gets hotter. P: Yes, its temperature rises very much. Why is this? L: During compression, the molecules of the air rub against each other, causing heat. P: This does indeed happen, but air is not very viscous. Therefore, it is highly unlikely, that such a strong increase of temperature would happen. L: In that case it must be that the heat of the air is compressed into a smaller space”. 

This is exactly how the researchers of the late 18th and early 19th century explained the phenomenon (cf. Section 3.3). First year engineering students who have developed a qualitative grasp of the *EQH* do much better when asked why the temperature rises in adiabatic compression ([175], p. 6) than if exposed to TET [172]. After a short introduction that includes demonstration and discussion of adiabatic change, about half of the students give correct explanations (in the sense of ECTH), and 90% can draw a proper *TS*-diagram. Importantly, for the case of isothermal expansion, which was not covered in the course, they can make largely correct inferences (90% say that the gas needs to be cooled—80% say entropy needs to be withdrawn; 70% sketch a proper *TS*-diagram and explain correctly why it takes this form).

In a formal course, after an understanding of the thermodynamics of an ideal gas, leading to expressions for the entropy capacitance (at constant volume), and the latent heat with respect to volume (Equations (10) and (11)) has been established, Equation (4) immediately leads to the differential equation for adiabatic processes. For instance, we can create the dynamical model for Rüchardt’s experiment (such as when a steel ball dropped into a thin vertical glass tube fitted on top of a larger container filled with air bounces up and down; see ([2],pp. 212,213); see also the more refined experimental setup presented in [176]). This allows one to study dissipative effects as well, particularly the effect of heat transfer between the wall of the vessel and the air, whose temperature goes up and down. It is fascinating to see that entropy transfer (and not just friction between the ball and the glass tube) leads to damping of the oscillatory motion.

### 5.2. Heat Engines and Minimization of Entropy Production

We now discuss approaches to thermal engines suitable for audiences ranging from laypersons to students of advanced engineering courses. When transformed into visual metaphoric representations, such as those afforded by process diagrams (Figure 2 and Figure 3), the schematic abstractions available to all of us let us produce useful descriptions of the operation and efficiency of thermal engines. Here, we restrict our discussion to heat engines where the fall of entropy makes energy available (Figure 3); we do not include heat pumps (but see [2], pp. 116, 174, and 177 for examples of process diagrams for heat pumps).

The starting point is the imagery—derived from Carnot’s waterfall model—of entropy falling through a temperature difference (Figure 3-1). As in the case of an actual waterfall, energy is made available by the fall of entropy at a rate determined solely by the strength of the flow and the potential difference through which the flow takes place. It simply does not matter what the energy made available will be used for. In the case of the waterfall, the water may wash out the rock over which it falls, dig a hole at the bottom, drive a water wheel, or all of these at once. In the case of the fall of entropy through a temperature difference, we may (ideally) have a situation where all the energy made available by the fall is used for driving an engine (Figure 3-2), or a situation of complete dissipation (Figure 3-3), or something in between that is more realistic for heat engines (Figure 3-4). No matter what follows, the starting point for discussing thermal engines is the expression for the power of a fall of entropy as expressed by Equation (9).

#### 5.2.1. Operation and Efficiency of an Ideal (Carnot) Heat Engine

We get an ideal (Carnot) heat engine [177] if 100% of the energy made available by the fall of entropy is “picked up” or used by the single (mechanical, electrical, …) process caused by the driving thermal process. In other words, the *naturally defined efficiency* [178] (i.e., useful power compared to available power), is 100% or 1.

When we study thermodynamics, we are commonly introduced to a different measure of efficiency, which is called thermal efficiency. It is the ratio of $I_{E3}/I_{E1}$ in Figure 3-2,-4, which is necessarily less than 1 since the entropy emitted by the (ideal) engine carries with it some energy according to Equation (18). The *Carnot efficiency* (i.e., the thermal efficiency of an ideal Carnot heat engine) is easily derived on the basis of entropy, energy balances, and Equation (18); on a single line of algebra, we get $\eta_{C}=\left(T_{1}-T_{2}\right)/T_{1}$ [179].

#### 5.2.2. Complete Dissipation in the Fall of Entropy

One case where a property of entropy may appear mystifying is the production of entropy in (conductive) heat transfer (Figure 3-3). Practices of calorimetry do not easily explain why heat transfer should be added to the list of irreversible processes; however, if we have access to an understanding of heat engines, we may gain insight into this phenomenon more easily. Typical irreversible processes such as friction, conductive flow of charge, and combustion, result from energy being made available and then (completely) dissipated. Since a fall of entropy from a high to low temperature likewise makes energy available, and since it is possible to place an engine producing “useful” power in this flow, we clearly have complete dissipation when no process producing this “useful” power is present—this is Carnot’s “true loss of motive power” (Section 3.4.3). With zero useful power, the efficiency of simple heat transfer is equal to 0.

Since the available power of a fall of entropy equals $\left(T_{1}-T_{2}\right)I_{S1}$, the rate of production of entropy (which sees the light of day at $T_{2}$) equals $\left(T_{1}-T_{2}\right)/T_{2}\cdot I_{S1}$. We get the same result if we apply entropy, energy balances, and Equation (18).

#### 5.2.3. Real Heat Engines and Optimization of Endoreversible Engines

In a real (i.e., irreversible) heat engine, part of the available power ($P_{th}$ in Figure 3-4) is used for producing entropy; this lowers the efficiency of the engine to values less than 1, and the same applies to the thermal efficiency (First Law efficiency). How much it is lowered depends upon the magnitude of the concrete irreversible processes at work.

There is a model of heat engines, called endoreversible, that limits irreversibility to that caused by heat transfer from the furnace to the working fluid and from the working fluid to the environment. If the furnace produces entropy at $T_{1}$, it first falls to $T_{1}^{*}$ (the upper operating temperature of the Carnot heat engine) through a first heat exchanger, dissipating all the energy made available in this fall. The entropy produced will raise the strength of the entropy current from $I_{S}$ to $I_{S}^{*}$. Then, entropy will be lowered reversibly in the working fluid to $T_{2}^{*}$
$T_{2}$, meaning that the useful power of the engine is reduced to $\left(T_{1}^{*}-T_{2}^{*}\right)I_{S}^{*}$. Finally, the current $I_{S}^{*}$ is transferred through a heat exchanger operating between $T_{2}^{*}$ and $T_{2}$ (the steady temperature of the environment), and additional entropy is produced.

It is possible to minimize the total entropy production rate obtained in this model [2], Section 9.2. The idea is that we should achieve maximum power if irreversibility is minimized (which cannot be equal to zero because then entropy will not flow from the furnace and into the environment—there needs to be a finite temperature difference for this to happen). The thermal efficiency of this model engine, first proposed by Curzon and Ahlborn [180] on the basis of maximizing useful power, is equal to $1-\sqrt{T_{2}/T_{1}}$ which is smaller than the Carnot efficiency. Interestingly, this models the efficiency of real power plants quite well.

Modeling thermal engines, both natural and engineered, as endoreversible, assumes that other dissipative processes can be neglected. This, plus minimization of entropy production rates, is a powerful tool for analyzing phenomena ranging from traditional thermal powerplants, to solar power plants, to thermal solar collectors, all the way to the “wind-engine” operating in our atmosphere ([2], Chapter 9). We may very well wonder why our atmosphere would “optimize” the output of this wind engine, but a simple model, which can also be applied to other planets, can give a useful estimate of how powerful the winds can actually be [181].

### 5.3. A Direct Entropic Approach to Thermoelectricity

Thermoelectricity is one of the phenomena considered to be “advanced” since it belongs to the realm of irreversible thermodynamics. If we take an approach based upon temperature and entropy as primitives (i.e., as thermal potential and *EQH*, respectively), a distinction between equilibrium and irreversible thermodynamics becomes meaningless—irreversibility and dynamical phenomena are always immediate ingredients of an ENT.

How to build a direct entropic approach to thermoelectricity has been demonstrated in [52], and a comparison of the entropy-form with the usual Onsager–de Groot–Callen model of irreversible thermodynamics is part of [159,160]. All we need, in addition to the usual expressions for conductive entropy and charge transports, and production rates of entropy due to these conductive flows, is an assumption that transports of electric charge and entropy are coupled directly, and to some degree, that the thermal tension produces an (thermo-)electric tension. The building blocks of this form of understanding follow directly from the phenomenology of the thermoelectric processes demonstrated with the help of simple experiments [52], pp. 256–257. Importantly, we do not need to first derive Onsager reciprocity relations in order to be able to formulate simple uniform or continuous dynamical models—the equality of Seebeck and Peltier coefficients follows naturally and very simply from considerations of available and useful power [52], p. 259.

Here is a sketch of the simplest (2-element) dynamical model of a thermoelectric device (see Figure 4). Imagine a thermoelectric device as consisting of two entropy storage elements (one each for the hot and cold sides) and two electric capacitors (one each for the electrically high and low sides). These storage elements are coupled internally and externally to thermal and electrical elements, respectively.

The coupling between the electric capacitors is well known from electric circuit models: here, we have a conductor and a thermoelectric generator element in series. In the thermal case, the connection between the hot and cold reservoirs is made by parallel conductive and thermoelectric elements—the thermoelectric entropy current, which isthe one coupled directly to charge flow, is in parallel to the conductive entropy flow caused by the temperature difference. The second element of thermoelectric coupling is the establishment of the thermoelectric tension in terms of the temperature difference. If we add rates of production of entropy due to the conductive transports of entropy and charge to the conductive entropy current, we have a complete dynamical model that, in all its simplicity, squares well with an observation of the dynamics of the device (cf. the observation made in [52], Figure 2). In summary, we have an approach to an example of irreversible thermodynamics suitable for introductory university physics and engineering courses.

### 5.4. Measuring Entropy as a High School Lab Activity

We would like to briefly outline how it is now possible to create a laboratory experiment for students focused on the measurement of the entropy change of a body being heated. We will follow Atkins’ ([182], Chapter 2) proposal—he stresses that one of the main difficulties we encounter in order to accept entropy as a useful and usable basic physical quantity is the lack of familiarity with the instrumentation used to measure it. From a conceptual point of view, a measuring device for the exchanged amounts of entropy should essentially consist of a probe we can put in contact with the body (imagined to be uniform) and an indicator that provides a numerical value. Here is a possible realization using data acquisition tools available today: we choose a body of water in a container (assumed to be perfectly insulating), inside which we place an electric heating element (which is properly put into operation by connecting it with a constant electric current power supply) and a thermometer (Figure 5, left). We then measure and record (a) the temperature, (b) the electric power, and (c) the production rate of entropy (according to Equation (19)) as functions of time. In addition, we let the data acquisition system calculate (d) the amount of energy supplied and (e) the entropy produced (and therefore, the entropy change) from the body as functions of time, starting from the moment the heater is turned on. All of this information can be displayed in real time on a computer screen (Figure 5, right).

Other data can be derived and displayed. In particular, if the electric power were kept constant, we would notice that as the temperature goes up, the entropy production rate decreases. We can create temperature - (specific) entropy (*T-s*) diagrams and derive the (specific) entropy capacity of a material [162]. Moreover, the experiment allows us to show to students how the entropy balance equation represents the link between the traditional definition of Clausius (involving reversible processes) and the procedure suggested by Atkins (using a totally dissipative process).

### 5.5. Naming the EQH

We have seen examples of how a direct approach to the *EQH* can be employed productively in the teaching and learning of thermodynamics. We have seen that this approach makes modeling of thermal processes in general, and thermal dynamical processes, much easier than what we are accustomed to from TET.

Let us assume that it is indeed simple and useful to introduce learners right from the start of a course on thermal physics to the idea of an *Extensive Quantity of Heat*, in a manner that is analogous to all the other fundamental extensive quantities operating in physical science [183,184,185,186,187]. This still leaves the question of how to deal with the naming of this quantity. Should we use (quantity or amount of) *heat* or *caloric* (see Table 2) or, as we have done in this and the previous section, *entropy*?

There is probably no single answer, and most likely neither is there a good answer, at least not for everyone and for every type of learning situation and level of formal sophistication. The dilemma of what to do is present in the work of Herrmann and co-workers who demonstrate that, quite obviously, *heat* is the term that best fits the concept of entropy [49], but nevertheless they prefer to use the word *entropy* in their middle and high school physics courses [95,184].

Our own experience with this matter can be summarized as follows. Using *entropy* as the name of the *EQH* in science courses for future kindergarten and primary school teachers [33,34] is a no-go. The word *entropy* could obviously never be used in these future teachers’ professional engagement with small children and young learners, which simply forbids us to take this route—these students need to learn how to create meaning and understanding using the cognitive tools of natural language and visual and embodied communication [188]. In an advanced high school course, on the other hand, we have been able to work with *entropy* as the *EQH*, and engender some useful understanding, especially regarding the operation of heat engines, in our students (cf. Section 5.2). In introductory courses designed for engineering students ([2], Chapters 1–9), we make the same observation; however, we also notice that students who have first been allowed to use the term *heat* for the *EQH*, and then are introduced to the term *entropy*, liberally confuse entropy, energy, and enthalpy for quite some time before arriving at a level and quality of understanding on par with what they exhibited when using only the term *heat*. There is certainly a high cognitive cost incurred with learning about the *EQH* if an artificially created term [189] such as *entropy* is used. 

## 6. Summary and Conclusions

Before we summarize the paper, we would like to make a comment regarding the scientific and philosophical status of an approach to thermal sciences that is not the *Traditional Equilibrium Thermodynamics* (TET) nor, for that matter, statistical physics. 

There are three or four distinct approaches to (or theories of) thermal phenomena: (1) traditional phenomenological thermodynamics; (2) microscopic approaches; (3) continuum thermodynamics. We may add to this (4) finite time thermodynamics. We can identify sub-categories such as (1a) Clausius’s and Kelvin’s forms, (1b) Gibbs’s rendering, (2a) kinetic theories, and (2b) statistical physics. There are “flavors” within (3) as well. Physics, chemistry, biology, applied physical science, and the technical sciences and engineering design make use of these forms to differing degrees.

We stress the word *distinct*: (1), (2), and (3) are *fundamentally distinct*, not just in form and range of application, but distinct from the viewpoints of natural philosophy, particularly epistemology, as it has been informed by modern cognitive science. Cognitively speaking, we are capable of taking different viewpoints (perspectives) that lead to *distinct* mental (imaginative, formal, theoretical…) structures (a perspective creates its own metaphoric web). *There is simply no way in which the results of one of these fundamental perspectives subsumes or even negates the products of one of the other viewpoints* as long as the models created within a framework reflect (a part of) reality. Each perspective can be relevant in its own way. Moreover, for very personal and practical reasons, everyone is allowed to prefer one perspective over another, but that does not negate the others.

In our paper, we avail ourselves of perspective (3) where entropy *is* the *extensive thermal quantity* (which applies to (1b) as well). This is our scientific basis which—as outlined above—should not be blended with other *distinct* perspectives or approaches.

Here is a short summary and outlook. We have made the point that thermal science, similarly to any other field of physics, needs to benefit from a direct approach to an *extensive quantity* (i.e., in addition to an *intensive* one, such as a *potential*) that is not to be confused with a quantity of energy, either transferred or stored. Despite the fact that the thermal extensive quantity, the *EQH*, has all but disappeared from *Traditional Equilibrium Thermodynamics* (TET), we believe that the experience of the *EQH* comes naturally to all of us—this is our reason for suggesting that we can have access to, and so formulate, if needed for educational or scientific reasons, an *Experientially Natural form of Thermodynamics* (ENT).

If we follow the course of historical development of thermal physics and accept basic abstract schemas as a foundation of understanding, we can create an *Extended Caloric Theory of Heat*, where caloric appears as a strongly generalized version of Clausius’ entropy—it most directly resembles the extended model of entropy as a fluidlike quantity known from continuum thermodynamics.

Applying such a direct approach to the *EQH* makes understanding and creation of (dynamical) models of thermal systems considerably easier and faster than if we apply a traditional approach known from TET. Importantly, we can use the same basic imaginative structures—those that make up our experience of *Forces of Nature*—when explaining thermal phenomena to laypersons, as they are used in advanced science and engineering courses. 

This still leaves open the question of how to name the *EQH*—there does not seem to be a simple, single, and uniquely applicable answer. We need to be aware that this question is quite important when we confront the challenge of how different communities—laypersons, journalists, learners of various ages in different educational settings, and scientists and engineers—could find common ground in their communication about thermal phenomena. All we can say is that we need to make every effort to create an ENT and develop a certain fluidity in the use of natural language when we speak about thermal systems and processes, a fluidity built upon the use of natural schematic, metaphoric, and analogical structures that are available to us all.

## Figures and Tables

**Figure 1 entropy-24-00646-f001:**
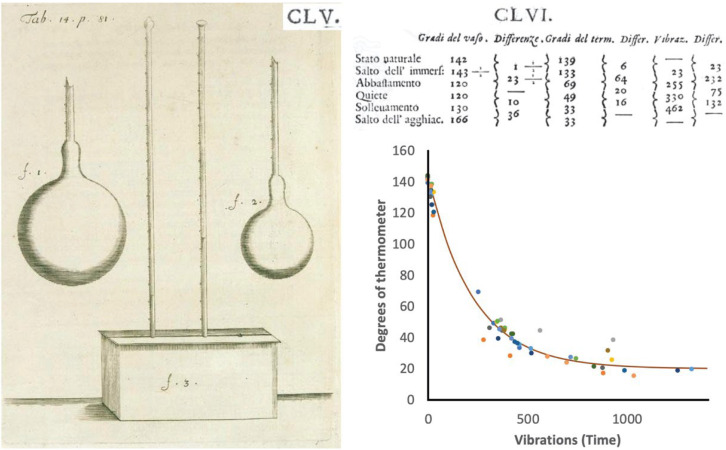
**Left**: experimental setup for measuring the artificial freezing of different liquids (the bulb on the left contains the liquid to be frozen; the bulb on the right containing alcohol is the Experimenters’ thermometer). **Right top**: an example of the data concerning the level of water in one of the bulbs and alcohol in the second (serving as the thermometer). **Right bottom**: diagram of the complete list of results reported for the level of alcohol in the neck of the second bulb, as a function of time (Vibrations), plus an exponentially decaying fitting curve (Source: [24], original page numbers included in the images).

**Figure 2 entropy-24-00646-f002:**
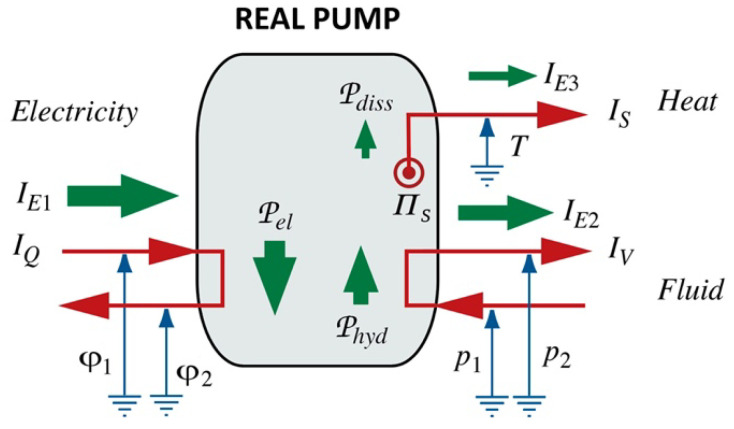
Process diagram for a real pump operating in steady-state, showing the action and interaction of forces of nature using visual metaphors for intensities (potentials, vertical levels), flows of extensive quantities (lines with arrows), production rate of entropy (circle with dot), power (vertical fat arrows inside rectangle; downward arrow symbolizes rate of energy made available, upward arrows denote energy used), and energy transfers (fat horizontal arrows). The gray rectangle denotes the pump as a background upon which the forces are active. (Source: Adapted from [2], Figure 2.7, p. 54).

**Figure 3 entropy-24-00646-f003:**
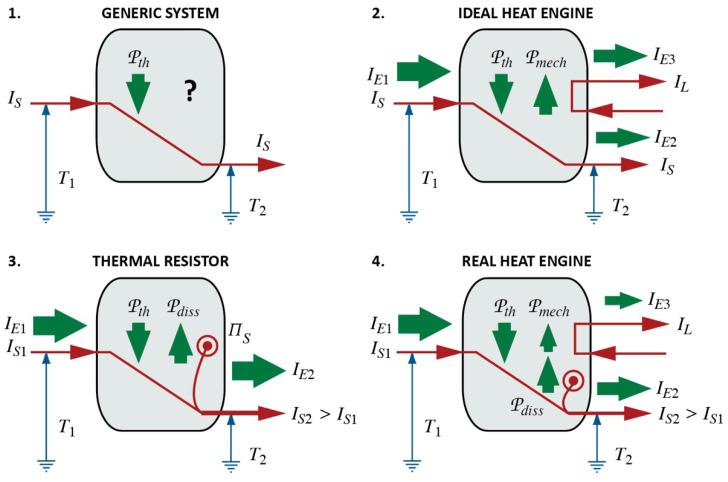
Process diagrams representing different (steady-state) processes caused by a fall of entropy through a temperature difference. **1**: “Generic waterfall image” of a causing thermal process where we leave open what might follow. **2**: Process diagram of an idea (Carnot) of a heat engine (an angular momentum current $I_{L}$ is “pumped” through its associated potential difference). **3**: Total dissipation in heat transfer through a thermal resistor. **4**: Real heat engine where energy made available is used for pumping angular momentum and producing entropy.

**Figure 4 entropy-24-00646-f004:**
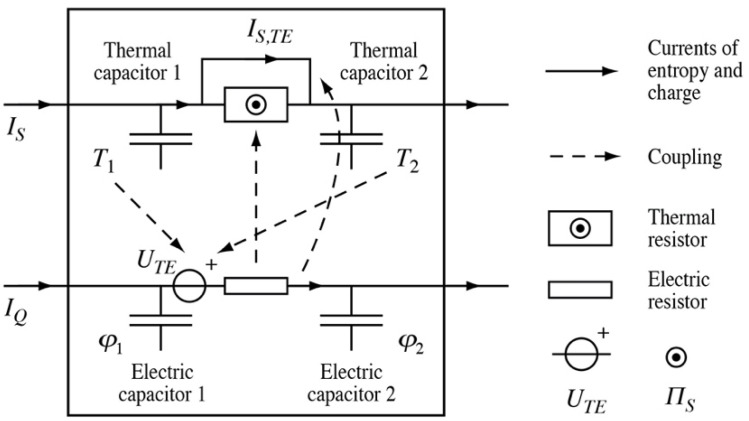
Simplest possible lumped parameter (uniform) dynamical model of a Peltier device. Thermally and electrically high and low faces are represented by capacitive elements. Transports of entropy ($I_{S}$, $I_{S,TE}$) and of charge ($I_{Q}$) take place between these capacitors. Entropy flows conductively, is carried by charge, and is produced ($\Pi_{S}$; source symbol at top center). A temperature difference sets up a thermoelectric voltage $U_{TE}$ (“electromotoric force”). Adapted from [52], Figure 5.

**Figure 5 entropy-24-00646-f005:**
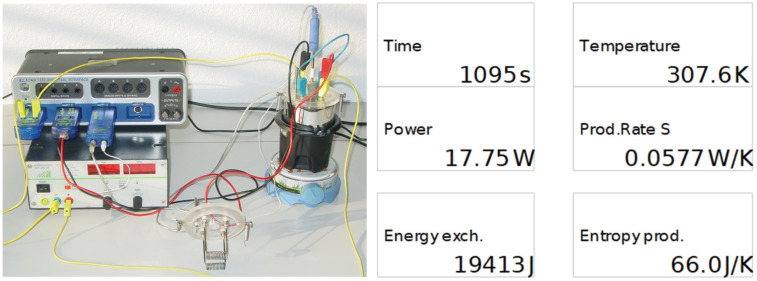
Measuring entropy quantities. Left: the experimental set-up. Right: A typical result.

**Table 1 entropy-24-00646-t001:** Abbreviations.

CP	Continuum Physics
CTH	Caloric Theory of Heat
DTH	Dynamical Theory of Heat
ECTH	Extended Caloric Theory of Heat
ENT	Experientially Natural form of Thermodynamics
*EQH*	Extensive Quantity of Heat
*ETQ*	Extensive Thermal Quantity
FoN	Force(s) of Nature
*LHHC*	Latent-Heat-and-Heat-Capacity Rule
TET	Traditional Equilibrium Thermodynamics
UDS	Uniform Dynamical Systems

**Table 2 entropy-24-00646-t002:** Symbols.

$\dot{x}$	Rate of change (derivative with respect to time)
C	Quantity of *EQH* (quantity of heat, caloric, thermal charge…)
$C_{abs}$	Quantity of *EQH* absorbed
$C_{em}$	Quantity of *EQH* emitted
I $,\;I_{C}$	Current, Current of *EQH*
$\Pi_{C}$	Production rate of *EQH*
$\Sigma_{C}$	Experientially Natural form of Thermodynamics
P	Power (*P_th_*: thermal power, *P_mech_*: (fluid) mechanical power)
E	Quantity of energy (energy of an element, energy stored)
$\;I_{E}$	Energy current
T	Temperature (ideal gas temperature, absolute temperature)
V	Volume
p	Pressure
n	Amount of substance
$\Lambda_{V}$	Latent heat (*EQH*, caloric…) with respect to volume
$\Lambda_{p}$	Latent heat (*EQH*, caloric…) with respect to pressure
$K_{V}$	Heat (*EQH*, caloric…) capacity at constant volume
$K_{p}$	Heat (*EQH*, caloric…) capacity at constant pressure
*R*	Universal gas constant
*γ*	$\mathrm{Ratio}\;\mathrm{of}\;\mathrm{heat}\;\mathrm{capacities}\;(K_{p}/K_{V}$)
$\lambda_{C}$	Conductivity with respect to *EQH* (quantity of heat, caloric…)
ρ	(Mass) density
κ	Specific (caloric) capacity
$F\left(T\right)$	Carnot’s function
$\left(\right){}^{\prime }$	Derivative with respect to temperature

## Data Availability

Not applicable.

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
