# Peer review of "Entropy and the Experience of Heat"

_entropy, 2022, doi:10.3390/e24050646_

Round 1

Reviewer 1 Report

The manuscript is written depending on the experiences of heat rather than the results from some experiments. It seems to be a popular science writing. I consider the authors had better present the method to measure the entropy, which could be related to “thermal stuff” mentioned by authors. Furthermore, the experimental results should be discussed to support the entropy relating to heat.

Reviewer 2 Report

It is not clear whether this is an English novel, or a social or cognitive science paper focussing on natural language understanding of the concept of heat; or an engineering paper submitted to a technical journal.

The description of entropy as extensive quantify of heat (EQH), relating EQH with forces of nature like electricity, motion and imagining entropy as kind of noon-conserved quasi-fluid, capable of being produced in irreversible process is thermodynamically flawed.

Discussion until Section is 3.3 is like reading a English literature based novel. Section 3.4 is interesting however, the idea of entropic current, description of fluidlike quantity of heat, the notion of ‘entropy fall’ with a waterfall analogy (Figure 3) along with equations described on lines 1375-1377 appears at best a novelist impression of scientific concepts. The authors do try to explain the entropy generation or irreversibility in during a simple heat conduction process with a loss of motive power. However, if Carnot work is produced when heat Q is being transferred across a temperature difference then only the difference will reach the lower temperature side. The analogy does not explain how the true loss of motive power will occur if all of Q needs to be transferred across the temperature difference rather than transferring just the difference.

Lines 1466-1481 do not make sense. Unfortunately, entropy does have many interpretations, and some them are actually wrong, but entropy defined as EQH does not make sense.

The authors are strongly advised to read Adrain Bejan’s paper on ‘Discipline in Thermodynamics’ published in the energies journal (doi:10.3390/en13102487) and Bejan’s work on quantifying irreversibility using entropy generation minimisation and then re-evaluate their strategy.

Reviewer 4 Report

Pleases ee the attached report.

Reviewer 5 Report

Dear Editors,

I have reviewed the manuscript Entropy and the Experience of Heat, which has been submitted by Fuchs, D’Anna, and Federico Corni for consideration for publication in Entropy.

The manuscript sets out to describe an Experientially Natural form of Thermodynamics (ENT), in which entropy is characterised as the central extensive quantity of heat. The argument is grounded in cognitive and educational research on our experiences and understanding of thermal phenomena, in particular of how we conceptualise heat as a Force of Nature. Another point of departure is the history of thermal science, in particular the assumptions behind and development of the caloric theory. The authors suggest that with certain modifications, such as avoiding connections to a particle theory of heat and allowing caloric to be a nonconserved quantity, many aspects of caloric theory can still be useful. An extended caloric theory offers an approach to the understanding and teaching thermal phenomena that is more in line with our experiences and intuitions than traditional approaches to thermodynamics. The authors show how this approach connects to Fuchs’ Dynamical Theory of Heat, with many references to examples in his textbook with the same name.

The study is well grounded in embodied cognitive science and educational research on students’ understanding of scientific concepts, in particular within thermal science. As the authors point out, the argument regarding the role of entropy as the central extensive quantity of heat aligns with Karlsruhe physics, which has received criticism for its unorthodox approach to physics teaching, but is therefore also interesting as an alternative in a situation where many students have difficulties understanding thermal concepts in traditional teaching. By the end, the authors point to the dilemma of what the extensive quantity of heat should be referred to in teaching. For students at younger ages, they call it heat, as it aligns with our intuitive understanding of heat. Higher up in the educational system, however, it becomes problematic to replace it with entropy, a notion that is not part of the everyday language.

I find the account of the historical development of the caloric theory valuable, including its basic assumptions, and discussion of the weaknesses that led to it being abandoned. In particular, the analysis of Carnot’s analogy between a waterfall and a heat engine is central in their description of the extensive quantity that is later identified as entropy. Rich quotes from referred texts are used as examples in a good way. I do not have the knowledge to assess the details of the historical description, but it is internally coherent and a central part of the line of argument.

Overall, I find the manuscript novel and well argued, and recommend that it can be published, provided minor revisions are made.

I have a few comments and suggestions for revision:

The references to Fuchs’ book Dynamical Theory of Heat seem to be wrong. I think [1] should be changed to [2] in the running text.

Line 272: I suggest omitting “Obviously”, and would like to see a motivation of in what sense these phenomena are Forces of Nature.

Line 421: “Note, these mappings do not represent metaphors; they are analogies.” I suggest emphasising the structural similarity between two domains in an analogy, in line with Gentner (rather than bidirectionality). There is often an asymmetry also in analogies, in projecting from the more known domain to the less known one.

Line 497: This translation sounds strange: “Around then the reason for the chill there have been variously speculated…”

Line 525: “Second, they did not confuse temperature and heat, intensity and extension, as has been claimed”. This should be motivated more clearly, or removed. What was wrong in the previous researchers’ conclusions?

Sections 3.3.3-3.3.4 I do not understand from the description how Laplace came to the conclusion that the speed of sound “is greater than the result derived by Newton by a factor equaling the square root of the ratio of the heat capacities at constant pressure and at constant volume”. I would like to see some more details on how it is motivated, qualitatively or quantitatively.

Line 694: Biot is misspelt.

Section 4. I suggest including more references to sections in Fuchs’ book Dynamical Theory of Heat where more details are given on the theory, in particular in Sections 4.2 and 4.3.

Section 4.4.3. The authors might consider referencing Kuhn for a critical view of the connection between Carnot’s caloric and entropy:

Kuhn, T. S. (1955). Carnot’s version of “Carnot's cycle”. American Journal of Physics, 23(2), 91-95.

Reference 53. The description of conceptual change research primarily accounts for research that takes for granted that “our “naïve” or common-sense concepts will certainly be wrong and misleading”. Rival theories that describe conceptual change in terms of restructuring of students’ existing productive resources (e.g. diSessa & Sherin 1998; Hammer, 2000), which seem to align more with the view of the authors, should be acknowledged:

diSessa, A. A., & Sherin, B. L. (1998). What changes in conceptual change? International Journal of Science Education, 20(10), 1155-1191.

Hammer, D. (2000). Student resources for learning introductory physics. American Journal of Physics, 68(S1), S52-S59.

Kind regards,

A reviewer

Round 2

Reviewer 4 Report

The author's responses are unconvincing. The authors  have not addressed satisfactorily the points raised in my previous report. The manuscript still has the same basic deficiencies mentioned in the report.  I regret to say that the manuscript can not be recommended for publication.

Reviewer 5 Report

Dear Editors,

I was the 5th reviewer in the previous round of reviews of the manuscript Entropy and the Experience of Heat, submitted by Fuchs et al.

The authors have addressed my comments and suggestions in a good way, and I recommend that the manuscript can be accepted for publication.

Kind regards,

Reviewer 5